# Factors related to mortality in patients with acute respiratory distress syndrome (ARDS) in a lower middle-income country: A retrospective observational study

Co Xuan Dao[1,2,3], Chinh Quoc Luong[2,3,4], Toshie Manabe[5,6], My Ha Nguyen[7], Dung Thi Pham[8], Quynh Thi Pham[2,9], Tai Thien Vu[2,10], Hau Thi Truong[1,2,4], Dai Quoc Khuong[2,11], Hien Duy Dang[1], Tuan Anh Nguyen[2,11], Thach The Pham[1,2,3], Giang Thi Huong Bui[1,2], Cuong Van Bui[2,3,12], Quan Huu Nguyen[2,3,11], Thong Huu Tran[2,3,11], Tan Cong Nguyen[1,2,3], Khoi Hong Vo[13,14,15], Lan Tuong Vu[2,11], Nga Thu Phan[7], Loc The Vu[8], Cuong Duy Nguyen[16], Thom Thi Vu[17], Anh Dat Nguyen[2,11], Chi Van Nguyen[2,11], Tuan Quoc Dang[1,2], Binh Gia Nguyen[1,18], Son Ngoc Do[1,2,3]*

1 Center for Critical Care Medicine, Bach Mai Hospital, Hanoi, Vietnam, 2 Department of Emergency and Critical Care Medicine, Hanoi Medical University, Hanoi, Vietnam, 3 Department of Emergency and Critical Care Medicine, Faculty of Medicine, VNU University of Medicine and Pharmacy, Vietnam National University, Hanoi, Vietnam, 4 Neuro Intensive Care Department, Neurology Center, Bach Mai Hospital, Hanoi, Vietnam, 5 Nagoya City University School of Data Science, Nagoya, Aichi, Japan, 6 Center for Clinical Research, Nagoya City University Hospital, Nagoya, Aichi, Japan, 7 Department of Health Organization and Management, Faculty of Public Health, Thai Binh University of Medicine and Pharmacy, Hung Yen, Vietnam, 8 Department of Nutrition and Food Safety, Faculty of Public Health, Thai Binh University of Medicine and Pharmacy, Hung Yen, Vietnam, 9 Intensive Care Unit, University Medical Center Ho Chi Minh City, University of Medicine and Pharmacy at Ho Chi Minh City, Ho Chi Minh City, Vietnam, 10 Emergency Department, Thai Nguyen National Hospital, Thai Nguyen, Vietnam, 11 Center for Emergency Medicine, Bach Mai Hospital, Hanoi, Vietnam, 12 Department of Intensive Care for Tropical Diseases, Bach Mai Institute for Tropical Medicine, Bach Mai Hospital, Hanoi, Vietnam, 13 Emergency Neurology Room, Neurology Center, Bach Mai Hospital, Hanoi, Vietnam, 14 Department of Neurology, Hanoi Medical University, Hanoi, Vietnam, 15 Department of Neurology, Faculty of Medicine, VNU University of Medicine and Pharmacy, Vietnam National University, Hanoi, Vietnam, 16 Department of Emergency and Critical Care Medicine, Thai Binh University of Medicine and Pharmacy, Hung Yen, Vietnam, 17 Department of Basic Medical Sciences, VNU University of Medicine and Pharmacy, Vietnam National University, Hanoi, Vietnam, 18 Department of Pre-Hospital Emergency Medicine, Faculty of Medicine, VNU University of Medicine and Pharmacy, Vietnam National University, Hanoi, Vietnam

* sonngocdo@gmail.com

## Abstract

### Background

Acute respiratory distress syndrome (ARDS) is associated with a high mortality rate, particularly in low- and middle-income countries, where the quality of pre-hospital or inter-hospital care can significantly impact patient outcomes. This study aimed to investigate mortality rates and associated factors among ARDS patients in Vietnam.

### Methods

This retrospective observational study included adult ARDS patients admitted to a central hospital in Vietnam from August 2015 to August 2023. Data was collected on

**Data availability statement:** All relevant data are within the manuscript and its Supporting information files.

**Funding:** The author(s) received no specific funding for this work.

**Competing interests:** The authors have declared that no competing interests exist.

inter-hospital care, patient characteristics, management, and outcomes; comparisons were made between survivors and non-survivors, and logistic regression analyses were performed to identify factors independently associated with hospital mortality.

## Results

Of 353 patients, 68.0% were male, the median age was 55.0 years (Q1-Q3: 39.0–66.0), and 61.5% died in the hospital. The majority of patients (89.5%; 316/353) were transferred from local hospitals, and 80.6% (253/314) had received non-invasive or invasive mechanical ventilation (MV) at the referring hospital. During transportation, 60.1% (116/193) had an endotracheal tube (ET) in place, and 25.6% (41/160) received non-invasive or invasive MV. Upon admission, the mean $PaO_2/FiO_2$ ratio was 110.04 mmHg (SD: 57.72), and the median Sequential Organ Failure Assessment (SOFA) score was 10.0 (Q1-Q3: 7.0–12.0). Most patients (95.7%; 315/329) received invasive MV on the first day of admission, and 36.7% (73/199) underwent cytokine adsorption during their hospital stay. The univariable logistic regression identified several factors significantly associated with hospital mortality, including age (OR: 1.027; 95% CI: 1.013–1.040; p < 0.001), $PaO_2/FiO_2$ ratio (OR: 0.993; 95% CI: 0.989–0.996; p < 0.001), SOFA Score (OR: 1.168; 95% CI: 1.093–1.250; p < 0.001), and septic shock (OR: 2.077; 95% CI: 1.338–3.226; p = 0.001). However, in multivariable analysis, only the use of an ET during transportation remained independently associated with reduced hospital mortality (adjusted OR: 0.070; 95% CI: 0.005–0.937; p = 0.045).

## Conclusions

This study investigated a selected cohort of patients and underscored the vital role of pre-hospital and inter-hospital care in ARDS outcomes in Vietnam. Most patients were transferred from local hospitals, with limited application of essential transport interventions such as ET and MV. Notably, the use of an ET during transfer was independently associated with reduced hospital mortality. To improve survival, healthcare strategies should prioritize strengthening inter-hospital transfer protocols, ensuring timely initiation of respiratory support, and expanding access to critical care resources across all levels of the healthcare system.

## Introduction

Acute respiratory distress syndrome (ARDS) is a form of respiratory failure characterized by the acute onset of bilateral alveolar opacities and hypoxemia [1], and it is associated with a variety of etiologies [2–6]. Despite advances in the care of critically ill patients, mortality rates for those with ARDS remain alarmingly high, ranging from 26% to 60% [6–11]. These rates vary across different institutions and countries, influenced by many confounding factors such as patient cohort characteristics, types

of mortality reported, comorbidities, treatment strategies, and disease severity. A multicenter, international cohort study of 3,022 patients with ARDS provides the best estimates, indicating an overall rate of death in the hospital of approximately 40.1% (952/2377) [9]. Mortality increases with disease severity; unadjusted hospital mortality was reported to be 34.9% (249/714) among those with mild ARDS, 40.3% (446/1106) for those with moderate disease, and 46.1% (257/557) for patients with severe ARDS [9]. Notably, the underlying root of ARDS is the leading cause of death among patients who die early [12–15]. In contrast, nosocomial pneumonia and sepsis are the most common causes of death among patients who die later in their clinical course [14]. Direct death from respiratory failure is uncommon [13]. Many studies have sought to identify factors during acute illness that predict mortality. Such factors can be categorized as patient-related [16–18], disease-related [19,20], and treatment-related [16,21,22]. However, no single factor has demonstrated clear superiority over others.

Vietnam is a lower middle-income country (LMIC), and is ranked 15th in the world and 3rd in Southeast Asia by population, with 96.462 million people. The country faces substantial challenges in providing care for critically ill patients. Vietnam has been a hotspot for emerging infectious diseases such as severe acute respiratory syndrome coronavirus 1 (SARS-CoV-1), avian influenza A(H5N1) [23], and coronavirus disease 2019 (COVID-19) [24–26], all of which have placed immense pressure on its healthcare infrastructure. Additionally, severe dengue, *Streptococcus suis* infections, malaria, and rising antibiotic resistance contribute to high sepsis rates in intensive care units (ICUs). As a result, pneumonia and sepsis are the leading causes of ARDS in Vietnam [25,27,28]. Despite rapid economic growth driven by economic and political reforms, Vietnam's healthcare system struggles with limited resources, insufficient access to advanced diagnostic and treatment strategies, and underdeveloped pre- and inter-hospital transfer systems [27,29,30]. These challenges are compounded by a shortage of trained medical staff capable of recognizing and effectively treating critically ill patients, including those with ARDS. Although national health insurance, established in 1992, aims to improve healthcare access and mitigate the impact of user fees, it does not fully cover the costs of advanced diagnostic and treatment options. As a result, central hospitals are often responsible for managing patients who cannot be adequately treated at local facilities [31], leading to delays in ARDS recognition, initiation of treatment, and delivery of appropriate supportive care.

Understanding country-specific etiologies, risk factors, and prognosis of ARDS is critical for reducing mortality in Vietnam and other countries with limited medical resources. Therefore, this study aimed to investigate the mortality rate and associated factors in patients with ARDS in Vietnam.

## Methods

### Study design and setting

This retrospective observational study is the major update of our previously published paper [27,32], which collected data on all ARDS patients admitted to the Bach Mai Hospital (BMH) in Hanoi, Vietnam, between August 2015 and August 2017 to elucidate the clinical epidemiology and disease prognosis in ARDS patients in Vietnam. To further investigate the mortality rate and associated factors from ARDS, especially those related to patient transportation, we continued to collect retrospective data on these patients admitted to the BMH between September 2017 and August 2023, following approval from the BMH Scientific and Ethics Committees (08/11/2023). The BMH is designated as a central general hospital with 3,200 beds in northern Vietnam by the Ministry of Health (MOH). In the healthcare system of Vietnam, the central hospitals (Level I) are responsible for training hospital staff and providing care for patients who cannot be adequately managed at lower-level facilities, including provincial and district hospitals (Levels II and III, according to the MOH of Vietnam) [31,33].

In Vietnam, patient transfers between medical facilities are governed by Circular No. 14/2014/TT-BYT, issued by the MOH of Vietnam on April 14, 2014 [33]. Circular No. 14/2014/TT-BYT outlines the procedures, conditions, and management related to patient transfers based on medical techniques. In emergencies, medical facilities must prepare for patient transport by ensuring: (i) the use of ambulances or other suitable vehicles; (ii) the availability of medical equipment and

emergency medicines; and (iii) assigning medical personnel, including doctors, nurses, and midwives, to supervise and appropriately care for the patient during transport. In this study, we categorized the type of inter-hospital transportation into four groups: (i) hospital ambulance services, denoting ambulances operated by hospitals; (ii) emergency medical services (EMS), indicating ambulances dispatched by an EMS dispatch center [30,34]; (iii) private ambulance services, describing ambulances that operate independently of an EMS dispatch center [30,34,35]; and (iv) own or private transport, or public transport [30,34]. We defined own or private transport as transport in vehicles by family members, relatives, neighbors, or passers-by [30,34]. Public transport includes taxis, buses, or other types of public transport [30,34]. Hospital ambulances and emergency medical services are staffed by trained and accredited medical personnel. These services are equipped with life-saving equipment, operate under medical oversight, and have regularly monitored quality indicators. In contrast, private ambulance services provide emergency transportation with limited medical interventions during transit. Patients arriving by private or public transport typically receive no first aid before reaching the hospital.

## Participants

This study included all patients aged 18 years or older who received a diagnosis of ARDS at BMH. The ARDS diagnosis was established by expert clinicians at the study site. The diagnosis of ARDS was based on the Berlin criteria [19,36], which encompasses: (i) *Timing*: The onset should occur within one week following a recognized clinical insult or the emergence or aggravation of respiratory symptoms; (ii) *Chest imaging*: The presence of bilateral opacities on imaging that cannot be entirely attributed to effusions, collapse of a lobe or the entire lung, or nodules; (iii) *Edema origin*: The respiratory failure should not be completely explainable by cardiac failure or excess fluid. An objective evaluation, such as echocardiography, is required to rule out hydrostatic edema in the absence of any risk factors; and (iv) *Oxygenation*: The oxygenation criterion is the ratio of partial pressure of oxygen in arterial blood ($PaO_2$ in mmHg) to the fraction of inspiratory oxygen concentration ($FiO_2$ expressed as a fraction, not a percentage) ($PaO_2/FiO_2$) ≤ 300 mmHg, with the patient receiving a minimum of 5 $cmH_2O$ of positive end-expiratory pressure (PEEP).

## Data collection

We used a standardized case record form (CRF) to extract data on relevant variables from patient medical records. Data was entered into the study database using EpiData Entry software (EpiData Association; Denmark, Europe), which enabled both simple and programmed entry functions to minimize errors and ensure data quality. Patient identifiers were excluded from the database to maintain confidentiality.

## Variables

The CRF contained four sections, which included variables mainly based on the Evidence-Based Clinical Practice Guidelines on the Use of Mechanical Ventilation (MV) in Adult Patients with Acute Respiratory Distress Syndrome (ARDS) [37] and collected by fully trained clinicians, such as information on:

(i)   The first section focused on baseline characteristics, such as inter-hospital care (e.g., prior hospitalization, inter-hospital transportation, inter-hospital care provider, inter-hospital airway, and respiratory care during the transport), demographics (i.e., age and gender), documented comorbidities (e.g., cerebrovascular disease, chronic cardiac failure, coronary artery disease (CAD)/myocardial infarction (MI), hypertension, chronic obstructive pulmonary disease (COPD)/asthma, other chronic pulmonary diseases (CPD), tuberculosis, active neoplasm, chronic renal failure, ulcer disease, diabetes mellitus, immunodeficiency, and hematological disease), and details of admission. We also used the 19 comorbidity categories to compute the Charlson Comorbidity Index (CCI) score, which measures the predicted mortality rate based on the presence of comorbidities [38]. A score of zero indicates that no comorbidities were detected; the higher the score, the higher the expected mortality rate is [38–40].

(ii) The second section comprised characteristics upon admission, such as vital signs (e.g., heart rate, respiration rate, arterial blood pressure, and body temperature), laboratory parameters (e.g., white blood cells [WBCs], hemoglobin, platelet, C-reactive protein [CRP], urea, creatinine, glucose, albumin, ferritin, and interleukin-6 [IL-6]), chest X-ray (CXR) findings (e.g., bilateral opacities and number of involved quadrants), gas exchange (e.g., $PaO_2$, partial pressure of carbon dioxide in the arterial blood [$PaCO_2$], the acidity of the blood [pH], and $PaO_2/FiO_2$ ratio), severity scoring systems, and respiratory pathogens (e.g., influenza A (H1N1) virus, cytomegalovirus [CMV], influenza B virus, and parainfluenza virus). The applied severity scoring systems in the study site include the Berlin criteria and Sequential Organ Failure Assessment (SOFA) Score. The Berlin criteria categorize ARDS severity based on the $PaO_2/FiO_2$ ratio as follows: mild ($200 < PaO_2/FiO_2 \leq 300$ mmHg), moderate ($100 < PaO_2/FiO_2 \leq 200$ mmHg), and severe ($PaO_2/FiO_2 \leq 100$ mmHg), with a minimum PEEP of 5 cmH$_2$O applied to the lungs at the end of each breath [19,36]. The SOFA score is a scoring system used to evaluate the severity of organ dysfunction and predict mortality in critically ill patients [41]. It encompasses six critical components related to the respiratory system (e.g., $PaO_2/FiO_2$ ratio), cardiovascular system (e.g., amount of vasoactive medication necessary to prevent hypotension), hepatic system (e.g., serum bilirubin level), coagulation system (e.g., platelet concentration), neurologic system (e.g., Glasgow coma score), and renal system (e.g., serum creatinine or urine output). Each system is assigned a score ranging from 0 (indicating normal function) to 4 (indicating severe abnormalities). The overall SOFA score can vary from 0 (optimal) to 24 (most severe), with higher scores indicating more severe conditions [41]. Additionally, virus infection was confirmed by a positive real-time reverse transcription-polymerase chain reaction test using samples obtained from the respiratory tract by the nasopharyngeal swab, the throat or mouth swab, or the tracheal lavage fluid.

(iii) The third section captured life-sustaining treatments provided during the ICU stay, such as respiratory support on the first and third day of admission (e.g., oxygen supplements and MV) and adjunctive therapies (e.g., prone positioning, recruitment maneuvers, extracorporeal membrane oxygenation [ECMO], antiviral drugs, antibiotics, corticosteroids, heparin, antiplatelet drugs, novel oral anticoagulants, continuous sedation, continuous neuromuscular blocking agents [NMBAs], renal replacement therapy [RRT], extracorporeal cytokine adsorption therapy [ECAT], tracheostomy, and inhaled vasodilators).

(iv) The fourth section is concerned with complications (e.g., hospital-acquired pneumonia [HAP], secondary bacterial infections, septic shock, multi-organ failure, deep venous thrombosis [DVT], gastrointestinal bleeding, and pneumothorax/pneumomediastinum) and clinical outcomes (e.g., hospital mortality). HAP is defined as pneumonia that occurs 48 hours or more after admission and does not appear to be incubating at the time of admission [42]. Additionally, septic shock is identified as a clinical construct of sepsis characterized by persisting hypotension requiring vasopressors to maintain a mean arterial pressure ≥65 mmHg, along with a serum lactate level > 2 mmol/L (18 mg/dL) despite adequate volume resuscitation [43].

## Outcomes

The primary outcome measured was hospital mortality, defined as death from any cause during hospitalization. At this study site, the attending physician decides whether to transfer or discharge ARDS patients, considering bed availability and overall hospital capacity. We also examined secondary outcomes, including complications and hospital length of stay (LOS).

## Sample size

In this retrospective observational study, the primary outcome was hospital mortality. Therefore, we used the formula to find the minimum sample size for estimating a population proportion with a confidence level of 95%, a confidence interval

(margin of error) of ±5.21%, and an assumed population proportion of 47.8%, based on the hospital mortality rate (47.8%) reported in a previously published study [6]. As a result, our sample size should be at least 353 patients, which might be large enough to reflect a normal distribution.

$$n = \frac{z^2 \mathrm{x}\, \hat{p}\,(1-\hat{p})}{\varepsilon^2}$$

where:

z is the z score (z score for a 95% confidence level is 1.96)

$\varepsilon$ is the margin of error ($\varepsilon$ for a confidence interval of ± 5.21% is 0.0521)

$\hat{p}$ is the population proportion $\hat{p}$ for a population proportion of 47.8% is 0.478)

n is the sample size

## Statistical analyses

We conducted all statistical analyses using IBM° SPSS° Statistics 22.0 (IBM Corp., Armonk, NY, USA). Data was reported as counts (no.) and percentages (%) for categorical variables and as medians with interquartile ranges (Q1–Q3) or means with standard deviations (SDs) for continuous variables, depending on their distribution. To compare character-istics between patients who survived and those who died in the hospital, as well as between patients who presented to the hospital during the pre-pandemic period of COVID-19 (August 2015 to December 2019) and those who presented to the hospital during the pandemic period of COVID-19 (January 2020 to August 2023), we applied the Chi-squared test or Fisher's exact test for categorical variables and the Mann–Whitney U test, Kruskal–Wallis test, or one-way analysis of variance (ANOVA) for continuous variables, as appropriate, based on normality and group size.

To predict hospital mortality among ARDS patients upon admission, we plotted receiver operating characteristic (ROC) curves and calculated areas under the ROC curves (AUROC) to determine the discriminatory ability of the SOFA score. In this analysis, a higher score indicates a more positive test result. Additionally, we examined the $PaO_2/FiO_2$ ratio, where a lower ratio signifies a more positive test result. Additionally, the optimal cut-off value of each score or ratio was determined by ROC curve analysis and defined as the point with the maximum value of Youden's index (i.e., sensitivity + specificity – 1). Based on these cut-off values, we categorized patients into two severity groups: those with scores below the cut-off value and those with scores at or above the cut-off value.

To identify factors associated with hospital mortality, we employed a logistic regression analysis that incorporates both univariable and multivariable analysis. The univariable logistic regression model evaluated a comprehensive set of variables potentially influencing hospital mortality. These included inter-hospital care factors (e.g., MV applied in prior hospitalization, inter-hospital transport, inter-hospital care provider, inter-hospital airway, and inter-hospital oxygen), demographic characteristics (e.g., age, gender), habitual behaviors (e.g., smoking history), documented comorbidities (e.g., chronic cardiac failure, active neoplasm, chronic renal failure, hematological disease, and CCI Score), clinical and laboratory parameters, gas exchange metrics (e.g., $PaO_2/FiO_2$ ratio), chest X-ray (CXR) findings, severity of illness (e.g., SOFA Score or SOFA Score ≥ cut-off), adjunctive therapies (e.g., recruitment maneuvers, corticosteroids, ECAT, and tra-cheostomy), and complications (e.g., HAP, secondary bacterial infections, and septic shock). For the multivariable logistic regression model, we implemented a systematic variable selection process to ensure statistical robustness and clinical relevance while minimizing overfitting. We initially selected variables based on a univariable analysis threshold of P < 0.25 for comparing death and survival in the hospital. These included inter-hospital care factors (i.e., inter-hospital transport, inter-hospital care provider, inter-hospital airway, and inter-hospital oxygen), demographics (i.e., age), documented comor-bidities (i.e., active neoplasm, chronic renal failure), gas exchange (i.e., $PaO_2/FiO_2$ ratio), severity of illness (i.e., SOFA Score or SOFA Score ≥ cut-off), adjunctive therapies (i.e., tracheostomy), and complications (i.e., secondary bacterial

infections, septic shock). Additionally, variables deemed clinically essential based on prior literature and expert consensus were included, such as inter-hospital care (i.e., MV applied in prior hospitalization) [44], demographics (i.e., gender) [45], documented comorbidities (i.e., chronic cardiac failure, hematological disease, CCI Score) [46], adjunctive therapies (i.e., recruitment maneuvers, corticosteroids, ECAT) [47–49], and complications (i.e., HAP) [50]. The final multivariable model comprised inter-hospital care factors (i.e., MV applied in prior hospitalization, inter-hospital transport, inter-hospital care provider, inter-hospital airway, and inter-hospital oxygen), demographics (i.e., age, gender), comorbidities (i.e., chronic cardiac failure, active neoplasm, chronic renal failure, hematological disease, CCI Score), gas exchange (i.e., $PaO_2/FiO_2$ ratio), severity of illness (i.e., SOFA Score or SOFA Score ≥ cut-off), adjunctive therapies (i.e., recruitment maneuvers, corticosteroids, ECAT, tracheostomy), and complications (i.e., HAP, secondary bacterial infections, septic shock). We refined the multivariable model using a stepwise backward elimination approach, starting with all selected variables and sequentially removing those with the least statistical significance until all remaining predictors were independently associated with hospital mortality ($P < 0.05$). We adhered to the events per variable (EPV) guideline of 10–50 outcome events per predictor to ensure model stability [51–54]. We rigorously assessed model validity, which included evaluating multicollinearity using the variance inflation factor (VIF) analysis. VIF values below 5 indicate acceptable predictor independence. Goodness-of-fit was confirmed using log-likelihood, pseudo-R-squared, and the Hosmer-Lemeshow test, ensuring the model's adequacy and calibration. Findings were reported as odds ratios (ORs) with 95% confidence intervals (CIs) for the univariable model and adjusted odds ratios (AORs) with 95% CIs for the multivariable model.

All statistical tests were two-tailed, with a significance threshold of $P < 0.05$.

## Ethical issues

On November 8, 2023, the BMH Scientific and Ethics Committees approved this study (Approval number: 6576/QD–BM; research code: BM_2023_160). The study was conducted following the Declaration of Helsinki. The BMH Scientific and Ethics Committees waived the need for informed consent for this retrospective observational study. Public notification of this study was made by public posting, according to the Strengthening the Reporting of Observational Studies in Epidemiology (STROBE): Explanation and Elaboration – the STROBE Statement – Checklist of items that should be included in reports of cohort studies. All data analyses were based on datasets kept in password-protected systems, and all final presented data have been anonymized.

## Results

In our study, we entered data from 374 patients with ARDS into the database. However, we excluded five patients who were under the age of 18. Additionally, we removed twelve patients with missing data for most variables (3.3%; 12/369). Furthermore, four duplicate entries were also removed (1.1%; 4/369). As a result, our analyses included 353 eligible patients (Fig 1).

### Prior hospitalization and inter-hospital care

In this study, most patients were transferred from local hospitals (89.5%; 316/353) (Table 1). Among these patients, 80.6% (253/314) previously received non-invasive or invasive MV at the referring hospital (Table 1). The mean LOS in the local hospital and the mean duration of MV in the referring hospital was 2.73 days (SD: 3.46) and 1.45 days (SD: 1.44), respectively (Table 1). Of all the patients, 54.7% (117/214) were transported to the central hospital by EMS, while 14.0% (30/214) utilized private ambulances, and only 29.4% (63/214) were transferred by hospital ambulances (Table 1). During transportation, most patients were attended to by EMS staff (52.7%; 106/201) or hospital nurses (26.9%; 54/201) (Table 1). However, notable for 19.9% (40/201) received care from bystanders (Table 1). As a result, only 60.1% (116/193) of patients had an endotracheal tube (ET), and only 25.6% (41/160) received non-invasive or invasive MV during transport (S1 Table as shown in S1 File). We examined the differences in prior hospitalization and inter-hospital care between patients who

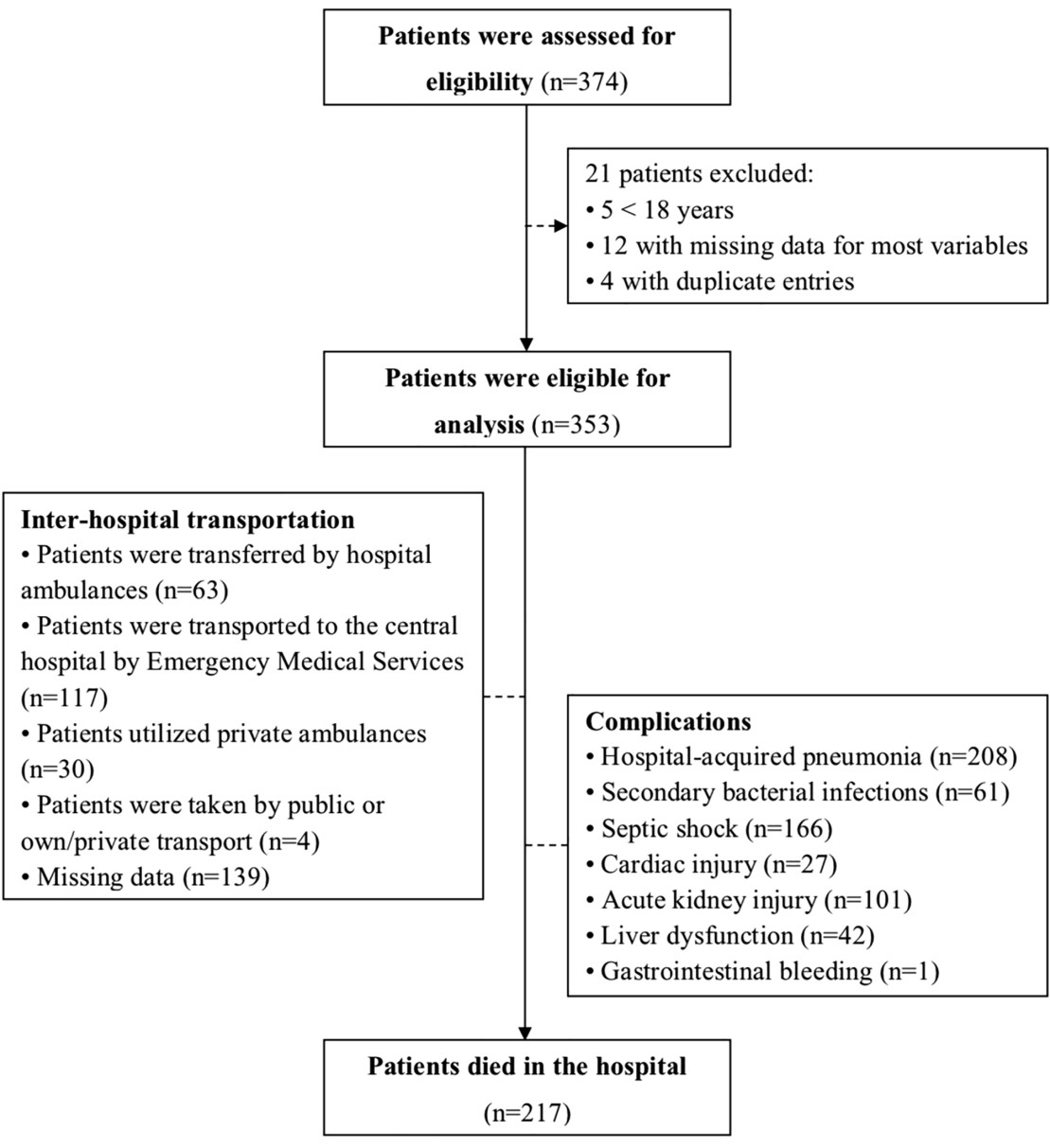

**Fig 1. Flowchart of the study design.**

survived and patients who died in the hospital, as shown in Table 1. Additionally, we analyzed these variables across various types of inter-hospital transportation, as detailed in S1 Table (S1 File).

## Clinical characteristics and management

In our study cohort, 68.0% (240/353) were male, and the median age was 55.0 years (Q1-Q3: 39.0–66.0; Table 2). Nearly half of the patients exhibited smoking habits, including those who had quit (14.3%, 31/217) and those who were current smokers (30.4%, 66/217; S2 Table as shown in S1 File). Commonly documented comorbidities included hypertension (29.5%, 67/227), diabetes mellitus (19.8%, 45/227), chronic renal failure (9.5%, 22/231), hematological diseases (7.3%,

**Table 1. Prior hospitalization and inter-hospital care for patients with acute respiratory distress syndrome, according to hospital survivability.**

| Variables | All cases n = 353 | Survived n = 136 | Died n = 217 | P value[a] |
|---|---|---|---|---|
| **Prior hospitalization** | | | | |
| Prior hospitalization, no. (%) | 316 (89.5) | 126 (92.6) | 190 (87.6) | 0.129 |
| Duration of stay (day), mean (SD), n = 313 | 2.73 (3.46) | 2.46 (3.31) | 2.90 (3.56) | 0.224 |
| MV applied[b], no. (%), n = 314 | 253 (80.6) | 102 (81.0) | 151 (80.3) | 0.889 |
| Duration of MV (day), mean (SD), n = 252 | 1.45 (1.44) | 1.43 (1.30) | 1.47 (1.53) | 0.766 |
| **Inter-hospital care** | | | | |
| The patient was brought in by, n = 214 | | | | 0.182 |
| EMS, no. (%) | 117 (54.7) | 37 (48.1) | 80 (58.4) | |
| Hospital ambulances, no. (%) | 63 (29.4) | 29 (37.7) | 34 (24.8) | |
| Private ambulances, no. (%) | 30 (14.0) | 9 (11.7) | 21 (15.3) | |
| Public or own/private transport, no. (%) | 4 (1.9) | 2 (2.6) | 2 (1.5) | |
| Inter-hospital care provider, n = 201 | | | | 0.389 |
| Bystanders[c], no. (%) | 40 (19.9) | 17 (23.6) | 23 (17.8) | |
| EMS staff, no. (%) | 106 (52.7) | 33 (45.8) | 73 (56.6) | |
| Hospital doctors, no. (%) | 1 (0.5) | 0 (0.0) | 1 (0.8) | |
| Hospital nurses, no. (%) | 54 (26.9) | 22 (30.6) | 32 (24.8) | |
| Inter-hospital airway | | | | |
| Endotracheal tube, no. (%), n = 193 | 116 (60.1) | 49 (67.1) | 67 (55.8) | 0.120 |

[a] To indicate comparisons between patients who survived and those who died in the hospital. [b] To indicate non-invasive or invasive MV at the referring hospital or during transportation. [c] To indicate family members, relatives, neighbors, layperson, police, or passers-by. Abbreviations: **EMS**, Emergency Medical Services; **MV**, mechanical ventilation.

17/234), and COPD/asthma (5.7%, 13/227; Table 2). The primary etiology of ARDS was pneumonia (91.1%, 205/225; Table 3), with the predominant pathogens being influenza A (H1N1) virus (12.8%, 29/227), CMV (2.7%; 6/221), influenza B virus (1.8%; 4/227), and parainfluenza virus (1.3%; 3/227; S3 Table as shown in S1 File). Among all patients, the mean $PaO_2/FiO_2$ ratio was 110.04 mmHg (SD: 57.72), with the $PaO_2/FiO_2$ ratio of ≤100 mmHg that was most commonly observed (56.0%, 196/350; Table 4). Additionally, the median SOFA Score was 10.0 (Q1-Q3: 7.0–12.0) upon admission (Table 4). Most patients (95.7%; 315/329) received invasive MV on the first day of admission (Table 5). Furthermore, 24.6% (85/345) of patients underwent prone positioning, and 17.2% (60/348) utilized recruitment maneuvers while on MV (Table 5). A majority of patients (100%; 138/138) received RRT, and 36.7% (73/199) underwent ECAT during the ICU stay (Table 5). We also conducted analyses to compare the clinical characteristics and management between patients who survived and those who died in the hospital, as shown in Tables 2–5 and S2–S5 (S1 File).

## Primary and secondary outcomes

Of the 353 eligible patients, 217 (61.5%) died in the hospital (Fig 1), and the mean LOS was 10.19 (SD: 11.54) days (Table 6). During the study, patients with ARDS experienced several common complications, including HAP (58.9%; 208/353), secondary bacterial infections (17.3%; 61/353), septic shock (47.0%; 166/353), acute kidney injury (28.6%; 101/353), and liver dysfunction (11.9%; 42/353; Table 4). Additionally, Table 6 compares complications between patients who survived and those who died in the hospital.

There were significant differences in patient variables between those admitted during the pre-pandemic period and those admitted during the pandemic period (Tables 7 and S6–S12 as shown in S1 File). For instance, the use of noninvasive or invasive MV at the referring hospital was significantly higher during the pre-pandemic period (97.9%, 139/142)

**Table 2. Demographics and comorbidities of patients with acute respiratory distress syndrome, according to hospital survivability.**

| Variables | All cases<br>n = 353 | Survived<br>n = 136 | Died<br>n = 217 | P value[a] |
|---|---|---|---|---|
| **Demographics** | | | | |
| Age (year), median (Q1-Q3) | 55.0 (39.0-66.0) | 44.5 (35.0-61.0) | 57.0 (43.0-69.0) | <0.001 |
| Gender (male), no. (%) | 240 (68.0) | 89 (65.4) | 151 (69.6) | 0.417 |
| **Comorbidities** | | | | |
| Cerebrovascular disease, no. (%), n = 351 | 7 (2.0) | 0 (0.0) | 7 (3.2) | 0.047 |
| Chronic cardiac failure, no. (%), n = 234 | 10 (4.3) | 2 (2.4) | 8 (5.3) | 0.502 |
| CAD/MI, no. (%), n = 222 | 4 (1.8) | 0 (0.0) | 4 (2.8) | 0.300 |
| Hypertension, no. (%), n = 227 | 67 (29.5) | 25 (30.1) | 42 (29.2) | 0.879 |
| COPD/asthma, no. (%), n = 227 | 13 (5.7) | 0 (0.0) | 13 (9.0) | 0.003 |
| Other CPD, no. (%), n = 347 | 27 (7.8) | 12 (9.0) | 15 (7.0) | 0.517 |
| Tuberculosis, no. (%), n = 227 | 5 (2.2) | 1 (1.2) | 4 (2.8) | 0.655 |
| Active neoplasm, no. (%), n = 227 | 11 (4.8) | 2 (2.4) | 9 (6.2) | 0.336 |
| Chronic renal failure, no. (%), n = 231 | 22 (9.5) | 11 (13.3) | 11 (7.4) | 0.148 |
| Ulcer disease, no. (%), n = 350 | 12 (3.4) | 4 (3.0) | 8 (3.7) | 0.773 |
| Diabetes mellitus, no. (%), n = 227 | 45 (19.8) | 16 (19.3) | 29 (20.1) | 0.875 |
| Immuno compromise, no. (%), n = 233 | 16 (6.9) | 7 (8.1) | 9 (6.1) | 0.557 |
| Hematological disease, no. (%), n = 234 | 17 (7.3) | 4 (4.7) | 13 (8.7) | 0.255 |
| CCI Score, median (Q1-Q3), n = 150 | 3.0 (1.0-5.0) | 2.0 (1.0-4.5) | 3.0 (1.0-5.0) | 0.310 |

[a] To indicate comparisons between patients who survived and those who died in the hospital. Abbreviations: **CAD**, coronary artery disease; **CCI**, Charlson Comorbidity Index; **COPD**, chronic obstructive pulmonary disease; **CPD**, chronic pulmonary disease; **MI**, myocardial infarction; **no.**, number of patients; **Q**, quartile.

compared to the pandemic period (66.3%, 114/172), with a p-value of < 0.001 (Table 7). Additionally, the rate of ET use during transport was also higher in the pandemic period (62.5%, 110/176) compared to the pre-pandemic period (35.3%, 6/17), with a p-value of 0.029 (Table 7). The prevalence of viral pneumonia, particularly due to the influenza A(H1N1) virus, was much greater in the pre-pandemic group (76.9%, 20/26) compared to the pandemic group (4.5%, 9/201), with a p-value of < 0.029 (Table 7). Furthermore, the use of invasive MV on the first day of admission was 100% (129/129) during the pre-pandemic period, while it was 93.0% (186/200) during the pandemic, with a p-value of 0.005 (Table 7). Despite these differences, the overall hospital mortality rates were not statistically significant, with rates of 59.9% (91/152) in the pre-pandemic group compared to 62.7% (126/201) in the pandemic group, resulting in a p-value of 0.590 (Table 7). Additional findings are available in S6–S12 Tables as shown in S1 File.

## Factors associated with hospital mortality

Although several factors were significantly associated with hospital mortality in the univariable logistic regression analyses, including age (OR: 1.027; 95% CI: 1.013–1.040; p < 0.001), $PaO_2/FiO_2$ ratio (OR: 0.993; 95% CI: 0.989–0.996; p < 0.001), SOFA score (OR: 1.168; 95% CI: 1.093–1.250; p < 0.001), tracheostomy (OR: 0.198; 95% CI: 0.062–0.632; p = 0.006), secondary bacterial infections (OR: 2.170; 95% CI: 1.159–4.064; p = 0.016), and septic shock (OR: 2.077; 95% CI: 1.338–3.226; p = 0.001), only the presence of an ET (AOR: 0.070; 95% CI: 0.005–0.937; p = 0.045) was identified in the multivariable logistic regression analysis as a protective factor for hospital mortality in patients with ARDS (Table 8). When we substituted the SOFA score with a threshold of ≥ 9.50 in the multivariable logistic regression analysis—based on the optimal cutoff value for distinguishing between patients who survived and those who died in the hospital (S13 Table in S1 File)—we identified additional independent predictors of hospital mortality (S14 Table in S1 File). These included a SOFA

**Table 3. Clinical and laboratory characteristics of patients with acute respiratory distress syndrome upon admission, according to hospital survivability.**

| Variables | All cases n = 353 | Survived n = 136 | Died n = 217 | P value[a] |
|---|---|---|---|---|
| **Etiology** | | | | |
| Etiology of ARDS, no. (%), n = 225 | | | | 0.019 |
| Pneumonia | 205 (91.1) | 69 (85.2) | 136 (94.4) | |
| Aspiration of gastric contents | 5 (2.2) | 1 (1.2) | 4 (2.8) | |
| Pulmonary contusion | 3 (1.3) | 2 (2.5) | 1 (0.7) | |
| Inhalation injury | 2 (0.9) | 1(1.2) | 1 (0.7) | |
| Pulmonary vasculitis | 1 (0.4) | 1 (1.2) | 0 (0.0) | |
| Drowning | 9 (4.0) | 7 (8.6) | 2 (1.4) | |
| **Clinical characteristics** | | | | |
| HR (beats/min), median (Q1-Q3), n = 227 | 123.0 (105.0-138.0) | 125.0 (105.0-135.0) | 123.0 (106.25-139.5) | 0.594 |
| RR (breaths/min), median (Q1-Q3), n = 227 | 27.0 (22.0-31.0) | 25.0 (20.0-30.0) | 28.0 (23.25-32.0) | 0.174 |
| Systolic BP (mmHg), mean (SD), n = 227 | 106.59 (24.75) | 111.30 (21.43) | 103.87 (26.16) | 0.025 |
| Diastolic BP (mmHg), mean (SD), n = 227 | 62.68 (15.22) | 66.57 (12.95) | 60.44 (16.01) | 0.006 |
| Body temperature (°C), mean (SD), n = 227 | 37.83 (1.04) | 37.68 (0.9) | 37.92 (1.11) | 0.167 |
| **Laboratory investigations** | | | | |
| WBCs (x10⁹/L), mean (SD), n = 352 | 14.73 (12.29) | 12.92 (7.71) | 15.87 (14.35) | 0.131 |
| Hemoglobin (g/L), mean (SD), n = 226 | 115.88 (25.15) | 115.06 (22.79) | 116.35 (26.49) | 0.722 |
| Platelet count (x10⁹/L), mean (SD), n = 352 | 178.13 (118.99) | 172.3 (97.8) | 181.81 (130.67) | 0.820 |
| CRP (mg/L), mean (SD), n = 84 | 63.11 (354.96) | 21.82 (24.04) | 86.05 (442.15) | 0.245 |
| Total bilirubin (μmol/L), mean (SD), n = 331 | 23.52 (35.24) | 27.82 (46.9) | 20.80 (25.03) | 0.702 |
| Glucose (mmol/L), mean (SD), n = 216 | 10.07 (5.80) | 9.39 (4.21) | 10.45 (6.52) | 0.476 |
| Ure (mmol/L), mean (SD), n = 226 | 11.98 (9.49) | 9.78 (7.47) | 13.26 (10.30) | 0.001 |
| Creatinine (μmol/L), mean (SD), n = 352 | 149.78 (147.5) | 138.35 (125.18) | 156.98 (159.82) | 0.143 |

[a] To indicate comparisons between patients who survived and those who died in the hospital. Abbreviations: **BP**, blood pressure; **CRP**, C-reactive protein; **HR**, heart rate; **no.**, number of patients; **Q**, quartile; **RR**, respiration rate; **SD**, standard deviation; **WBCs**, white blood cells.

Score of ≥9.50 (AOR: 14.819; 95% CI: 1.410–155.760; p = 0.025) and ECAT (AOR: 18.259; 95% CI: 1.038–321.089; p = 0.047).

## Discussion

In the present study, we found that over three-fifths of patients with ARDS died in the hospital (Fig 1). Most patients had received noninvasive or invasive MV at the referring hospital (Table 1). However, only over three-fifths of patients had an ET, and only over a fourth received noninvasive or invasive MV during transportation (S1 Table as shown in S1 File). On admission, the mean $PaO_2/FiO_2$ ratio was low, with the $PaO_2/FiO_2$ ratio of ≤100 mmHg being the most prevalent finding (Table 4). The most common etiology of ARDS was pneumonia (Table 3), most often caused by the influenza A (H1N1) virus (S3 Table in S1 File). As a result, the median SOFA score was high on admission (Table 4). Most patients received invasive MV on the first day of admission (Table 5). A substantial proportion of patients received RRT, and nearly two-fifths underwent ECAT during the ICU stay (Table 5). Common complications included HAP and septic shock (Table 6). Notably, ET use during transportation was independently associated with reduced hospital mortality (Tables 8 and S14 as shown in S1 File).

Data on ARDS in Vietnam remains scarce. However, the hospital mortality rate in this study is consistent with our prior report (57.1%; 72/126) [27], likely due to uniform inclusion criteria and the study setting within the same hospital

**Table 4. Arterial blood gas parameters, chest X-ray findings, and severity of illness upon admission of patients with acute respiratory distress syndrome, according to hospital survivability.**

| Variables | All cases n=353 | Survived n=136 | Died n=217 | P value[a] |
|---|---|---|---|---|
| **Arterial blood gas** | | | | |
| pH, mean (SD), n=350 | 7.32 (0.16) | 7.34 (0.12) | 7.30 (0.18) | 0.033 |
| $PaO_2$ (mmHg), mean (SD), n=349 | 81.29 (35.25) | 87.91 (40.83) | 77.06 (30.53) | 0.046 |
| $PaCO_2$ (mmHg), mean (SD), n=350 | 44.17 (16.1) | 43.46 (14.43) | 44.62 (17.09) | 0.750 |
| $SpO_2$ (%), mean (SD), n=348 | 92.24 (49.03) | 97.87 (77.61) | 88.63 (8.09) | <0.001 |
| $PaO_2/FiO_2$ ratio (mmHg), n=350 | 110.04 (57.72) | 125.09 (67.35) | 100.48 (48.45) | 0.004 |
| **Initial chest X-ray findings** | | | | |
| Bilateral opacities, no. (%), n=352 | 350 (99.4) | 136 (100.0) | 214 (99.1) | 0.524 |
| Number of involved quadrants, n=341 | | | | 0.223 |
| 1 quadrant, no. (%) | 5 (1.5) | 2 (1.5) | 3 (1.4) | |
| 2 quadrants, no. (%) | 10 (2.9) | 4 (3.0) | 6 (2.9) | |
| 3 quadrants, no. (%) | 88 (25.8) | 42 (31.8) | 46 (22.0) | |
| 4 quadrants, no. (%) | 238 (69.8) | 84 (63.6) | 154 (73.7) | |
| **Severity of illness** | | | | |
| Berlin criteria, no. (%), n=350 | | | | 0.001 |
| 200 mmHg< $PaO_2/FiO_2$ ≤300 mmHg | 33 (9.4) | 22 (16.2) | 11 (5.1) | |
| 100 mmHg< $PaO_2/FiO_2$ ≤200 mmHg | 121 (34.6) | 50 (36.8) | 71 (33.2) | |
| $PaO_2/FiO_2$ ≤100 mmHg | 196 (56.0) | 64 (47.1) | 132 (61.7) | |
| SOFA, median (Q1-Q3), n=335 | 10.0 (7.0-12.0) | 8.0 (6.0-11.0) | 10.0 (7.0-12.0) | <0.001 |

a) To indicate comparisons between patients who survived and those who died in the hospital. Abbreviations: **$PaCO_2$**, partial pressure of carbon dioxide in the arterial blood; **$PaO_2$**, partial pressure of oxygen in the arterial blood; **pH**, the acidity of the blood; **no.**, number of patients; **SD**, standard deviation; **SOFA Score**, Sequential Organ Failure Assessment Score; **$SpO_2$**, saturation of oxygen in the peripheral blood.

during a similar period. In contrast, our mortality rate exceeds those reported in international studies, including the United States Validation of Biomarkers in Acute Lung Injury Diagnosis (VALID) study (23.7%; 153/646) [8], the Spanish Acute Lung Injury: Epidemiology and Natural History (ALIEN) study (47.8%; 122/255) [6], and the Large Observational Study to Understand the Global Impact of Severe Acute Respiratory Failure (LUNG SAFE) study (40.1%; 952/2377) [9]. These disparities may reflect differences in patient characteristics, pathogen profiles, and critical care capacity between LMICs and HICs.

A key factor in our study is the selective nature of our cohort, as most ARDS patients were transferred from local hospitals to a central tertiary facility (Table 1), introducing significant referral bias. Patients are typically transferred only after initial management fails at lower-level facilities, resulting in a cohort with more advanced disease severity upon arrival. In Vietnam, patients with ARDS are often initially diagnosed with severe pneumonia at local hospitals and transferred only when their condition worsens, delaying diagnosis and treatment and potentially contributing to higher mortality [26,27,31,32]. Resource limitations further compound this issue. Vietnam's healthcare system faces chronic underfunding, shortages of trained intensivists, advanced diagnostic equipment, and critical care beds, particularly at provincial and district levels (levels II and III per the MOH) [29]. These constraints lead to suboptimal early interventions, such as delayed initiation of lung-protective ventilation or adjunctive therapies, which are more readily available in HICs.

Pneumonia was the predominant etiology of ARDS in our study (Table 3), aligning with prior studies such as VALID (19.3%; 125/646) [8], ALIEN (42.4%; 108/255) [6], and LUNG SAFE (59.4%; 1794/3022) [9], though our cohort exhibited a higher prevalence. This predominance likely exacerbates mortality, as pneumonia and sepsis are leading causes of

**Table 5. Management strategies for patients with acute respiratory distress syndrome, according to hospital survivability.**

| Variables | All cases<br>n = 353 | Survived<br>n = 136 | Died<br>n = 217 | P value[a] |
|---|---|---|---|---|
| **Respiratory support** | | | | |
| The first day respiratory support, n = 329 | | | | 0.916 |
| Oxygen only, no. (%) | 7 (2.1) | 3 (2.4) | 4 (2.0) | |
| Non-invasive MV, no. (%) | 7 (2.1) | 2 (1.6) | 5 (2.5) | |
| Invasive MV, no. (%) | 315 (95.7) | 122 (96.1) | 193 (95.5) | |
| None of the above, no. (%) | 0 (0.0) | 0 (0.0) | 0 (0.0) | |
| **Adjunctive therapies during ICU stay** | | | | |
| Prone positioning, no. (%), n = 345 | 85 (24.6) | 26 (19.7) | 59 (27.7) | 0.094 |
| Recruitment maneuvers, no. (%), n = 348 | 60 (17.2) | 20 (14.9) | 40 (18.7) | 0.365 |
| ECMO, no. (%), n = 263 | 22 (8.4) | 9 (8.7) | 13 (8.2) | 0.891 |
| Antiviral drugs, no. (%), n = 335 | 70 (20.9) | 39 (29.5) | 31 (15.3) | 0.002 |
| Antibiotics, no. (%), n = 335 | 328 (97.9) | 130 (98.5) | 198 (97.5) | 0.708 |
| Corticosteroids, no. (%), n = 342 | 82 (24.0) | 33 (25.4) | 49 (23.1) | 0.633 |
| Continuous sedation, no. (%), n = 349 | 333 (95.4) | 128 (96.2) | 205 (94.9) | 0.563 |
| NMBAs, no. (%), n = 345 | 247 (71.6) | 88 (67.2) | 159 (74.3) | 0.154 |
| RRT, no. (%), n = 138 | 138 (100.0) | 46 (100.0) | 92 (100.0) | NA |
| ECAT, no. (%), n = 199 | 73 (36.7) | 28 (37.8) | 45 (36.0) | 0.795 |
| Tracheostomy, no. (%), n = 263 | 16 (6.1) | 12 (11.5) | 4 (2.5) | 0.003 |
| Inhaled vasodilators, no. (%), n = 227 | 1 (0.4) | 1 (1.2) | 0 (0.0) | 0.366* |
| Neutrophil elastase therapy, no. (%), n = 226 | 1 (0.4) | 1 (1.2) | 0 (0.0) | 0.363* |

a) To indicate comparisons between patients who survived and those who died in the hospital. Abbreviations: **ECAT**, extracorporeal cytokine adsorption therapy; **ECMO**, extracorporeal membrane oxygenation; **MV**, mechanical ventilation; **NA**, not available; **NMBAs**, neuromuscular blocking agents; **no.**, number of patients; **RRT**, renal replacement therapy; **SD**, standard deviation.

**Table 6. Clinical time course and complications in patients with acute respiratory distress syndrome, according to hospital survivability.**

| Variables | All cases<br>n = 353 | Survived<br>n = 136 | Died<br>n = 217 | P value[a] |
|---|---|---|---|---|
| **Clinical time-course** | | | | |
| LOS (day), mean (SD), n = 348 | 10.19 (11.54) | 15.33 (13.02) | 7.01 (9.20) | <0.001 |
| **Complications** | | | | |
| HAP, no. (%) | 208 (58.9) | 79 (58.1) | 129 (59.4) | 0.801 |
| Secondary bacterial infections, no. (%) | 61 (17.3) | 15 (11.0) | 46 (21.2) | 0.014 |
| Septic shock, no. (%) | 166 (47.0) | 49 (36.0) | 117 (53.9) | 0.001 |
| Cardiac injury, no. (%) | 27 (7.6) | 9 (6.6) | 18 (8.3) | 0.564 |
| Acute kidney injury, no. (%) | 101 (28.6) | 30 (22.1) | 71 (32.7) | 0.031 |
| Liver dysfunction, no. (%) | 42 (11.9) | 15 (11.0) | 27 (12.4) | 0.690 |
| Gastrointestinal bleeding, no. (%) | 1 (0.3) | 1 (0.7) | 0 (0.0) | 0.385 |

a) To indicate comparisons between patients who survived and those who died in the hospital. Abbreviations: **HAP**, hospital-acquired pneumonia; **ICU**, intensive care unit; **LOS**, hospital lengths of stay; **no.**, number of patients; **SD**, standard deviation.

**Table 7. Comparative analysis of prior hospitalization, inter-hospital care, respiratory viral pathogens, respiratory support, and outcome in patients with acute respiratory distress syndrome before and during the COVID-19 pandemic.**

| Variables | All cases n=353 | Aug 2015 – Dec 2019 n=152 | Jan 2020 – Aug 2023 n=201 | P value[a] |
|---|---|---|---|---|
| **Prior hospitalization** | | | | |
| Prior hospitalization, no. (%) | 316 (89.5) | 144 (94.7) | 172 (85.6) | 0.005 |
| Duration of stay (day), mean (SD), n=313 | 2.73 (3.46) | 1.43 (2.02) | 3.79 (4.0) | <0.001 |
| MV applied[b], no. (%), n=314 | 253 (80.6) | 139 (97.9) | 114 (66.3) | <0.001 |
| Duration of MV (day), mean (SD), n=252 | 1.45 (1.44) | 1.09 (0.29) | 1.89 (2.05) | <0.001 |
| **Inter-hospital care** | | | | |
| Inter-hospital airway | | | | |
| Endotracheal tube, no. (%), n=193 | 116 (60.1) | 6 (35.3) | 110 (62.5) | 0.029 |
| **Respiratory viral pathogens** | | | | |
| Influenza virus A (H1N1), no. (%), n=227 | 29 (12.8) | 20 (76.9) | 9 (4.5) | <0.001 |
| **Respiratory support** | | | | |
| The first day respiratory support, n=329 | | | | 0.005 |
| Oxygen only, no. (%) | 7 (2.1) | 0 (0.0) | 7 (3.5) | |
| Non-invasive MV, no. (%) | 7 (2.1) | 0 (0.0) | 7 (3.5) | |
| Invasive MV, no. (%) | 315 (95.7) | 129 (100.0) | 186 (93.0) | |
| **Outcome** | | | | |
| Died in the hospital | 217 (61.5) | 91 (59.9) | 126 (62.7) | 0.590 |

a)To indicate comparisons between patients who presented to the hospital during the pre-pandemic period of COVID-19 (August 2015 to December 2019) and those who presented to the hospital during the pandemic period of COVID-19 (January 2020 to August 2023). b) To indicate non-invasive or invasive MV at the referring hospital or during transportation. Abbreviations: **MV**, mechanical ventilation; **no.**, number of patients; **SD**, standard deviation.

early death in ARDS, alongside multiple organ dysfunction syndrome (MODS) [12–15]. In the present study, the SOFA score (Table 4) is comparable to those in LUNG SAFE (mean 10.1 [95% CI: 9.9–10.2]) [9], but our $PaO_2/FiO_2$ ratio is lower (Table 4 vs. 161 mmHg [95% CI: 158–163] in LUNG SAFE) [9], suggesting greater hypoxemia severity. A large observational study (n=3670) also indicates that mortality rises with ARDS severity [19,36], potentially explaining our elevated mortality rates due to the high pneumonia burden. Additionally, the lack of widespread access to advanced therapies, such as ECMO (utilized in only 8.5% of our cohort; Table 5) and high-flow nasal cannula (HFNC) during transport or early management [29,30,55], limits rescue options for refractory cases. In contrast to HICs, where ECMO is more routinely available and associated with better outcomes in high-volume centers [56,57], LMICs like Vietnam face barriers including high costs, limited expertise, infrastructure gaps, and global disparities in access. These factors collectively contribute to the higher mortality observed here compared to international cohorts (e.g., 23.7% in VALID [8] and 40.1% in LUNG SAFE [9]). To reduce ARDS mortality in Vietnam, enhancing local healthcare resources (e.g., human expertise, medical equipment, and infrastructure) is critical, as evidenced by ongoing efforts to upgrade hospitals and address workforce shortages.

Patient transfers from local to central hospitals may further worsen outcomes. Most patients in our study were transferred without ET or MV (Tables 1 and S1 as shown in S1 File), and limited ET use during transport was independently associated with increased hospital mortality (Tables 8 and S14 in S1 File). A German prospective cohort study demonstrated that mechanically ventilated patients with ARDS experienced fewer transport-related complications, highlighting the importance of optimized inter-hospital transfer protocols [58]. Similarly, a United States observational study found that delayed intubation was associated with higher mortality rates (50.0% [18/36] vs. 29.6% [104/351] for early intubation and 14.3% [10/70] for patients never requiring intubation) [22]. These findings suggest that early interventions, such as HFNC therapy, non-invasive ventilation, or intubation, may influence outcomes [22,59]. Although our study lacks data on HFNC

**Table 8. Factors associated with hospital mortality in patients with acute respiratory distress syndrome.**

| Factors | Univariable logistic regression analyses[a] | | | | Multivariable logistic regression analysis[b] | | | |
|---|---|---|---|---|---|---|---|---|
| | OR | 95% CI for OR | | p-value | AOR | 95% CI for AOR | | p-value |
| | | Lower | Upper | | | Lower | Upper | |
| **Prior hospitalization** | | | | | | | | |
| MV[c] | 0.960 | 0.542 | 1.701 | 0.889 | NA | NA | NA | NA |
| **Inter-hospital care** | | | | | | | | |
| The patient was brought in by | | | | | | | | |
| EMS | Reference | | | 0.215 | NA | | | NA |
| Hospital ambulances | 0.542 | 0.289 | 1.018 | 0.057 | NA | NA | NA | NA |
| Private ambulances | 1.079 | 0.451 | 2.583 | 0.864 | NA | NA | NA | NA |
| Inter-hospital care provider | | | | | | | | |
| Bystanders[d] | Reference | | | 0.505 | NA | | | NA |
| EMS staff | 1.635 | 0.773 | 3.460 | 0.199 | NA | NA | NA | NA |
| Hospital nurses | 1.075 | 0.469 | 2.464 | 0.864 | NA | NA | NA | NA |
| Inter-hospital airway | | | | | | | | |
| Endotracheal tube | 0.619 | 0.337 | 1.136 | 0.122 | 0.070 | 0.005 | 0.937 | 0.045 |
| Inter-hospital oxygen | | | | | | | | |
| Nasal cannula | Reference | | | 0.429 | NA | | | NA |
| Facial mask | 0.867 | 0.241 | 3.110 | 0.826 | NA | NA | NA | NA |
| Bag valve mask | 0.292 | 0.082 | 1.033 | 0.056 | NA | NA | NA | NA |
| Mechanical ventilator[c] | 0.386 | 0.146 | 1.021 | 0.055 | NA | NA | NA | NA |
| Others[e] | 0.576 | 0.191 | 1.679 | 0.305 | NA | NA | NA | NA |
| None of the above | 0.542 | 0.170 | 1.727 | 0.300 | NA | NA | NA | NA |
| **Demographics** | | | | | | | | |
| Age (year) | 1.027 | 1.013 | 1.040 | <0.001 | 1.056 | 0.981 | 1.136 | 0.150 |
| Gender (male) | 0.828 | 0.524 | 1.307 | 0.417 | NA | NA | NA | NA |
| **Documented Comorbidities** | | | | | | | | |
| Chronic cardiac failure | 2.310 | 0.479 | 11.138 | 0.297 | NA | NA | NA | NA |
| Active neoplasm | 2.700 | 0.569 | 12.807 | 0.221 | NA | NA | NA | NA |
| Chronic renal failure | 0.526 | 0.217 | 1.271 | 0.153 | NA | NA | NA | NA |
| Hematological disease | 1.936 | 0.611 | 6.137 | 0.262 | NA | NA | NA | NA |
| CCI Score | 1.086 | 0.914 | 1.290 | 0.347 | 0.591 | 0.278 | 1.254 | 0.170 |
| **Severity of illness** | | | | | | | | |
| PaO$_2$/FiO$_2$ ratio | 0.993 | 0.989 | 0.996 | <0.001 | NA | NA | NA | NA |
| SOFA Score | 1.168 | 1.093 | 1.250 | <0.001 | 1.362 | 0.960 | 1.932 | 0.083 |
| **Adjunctive therapies** | | | | | | | | |
| Recruitment maneuver | 1.310 | 0.729 | 2.355 | 0.366 | NA | NA | NA | NA |
| Corticosteroid | 0.884 | 0.532 | 1.468 | 0.633 | NA | NA | NA | NA |
| ECAT | 0.924 | 0.510 | 1.676 | 0.795 | 8.365 | 0.569 | 122.964 | 0.121 |
| Tracheostomy | 0.198 | 0.062 | 0.632 | 0.006 | NA | NA | NA | NA |
| **Complications** | | | | | | | | |
| HAP | 1.058 | 0.684 | 1.635 | 0.801 | NA | NA | NA | NA |
| Secondary bacterial infections | 2.170 | 1.159 | 4.064 | 0.016 | NA | NA | NA | NA |
| Septic shock | 2.077 | 1.338 | 3.226 | 0.001 | 5.721 | 0.410 | 79.745 | 0.194 |
| Constant | | | | | 0.033 | | | 0.182 |

a) Each variable of the inter-hospital care, demographics, documented comorbidities, clinical and laboratory features, gas exchange, chest X-ray findings, severity of illness, oxygen supplement, MV, adjunctive therapies, and complications was analyzed in the univariable logistic regression model and was

*(Continued)*

**Table 8.** (Continued)

considered in the multivariable logistic regression model if the P-value was < 0.25 in univariable logistic regression analysis, as well as clinically crucial factors; [b] All selected variables were included in the multivariable logistic regression model with the stepwise backward elimination method. Variables, then, were deleted stepwise from the full model until all remaining variables were independently associated with hospital mortality. To ensure the robustness of our model, we evaluated multicollinearity among the predictor variables using the variance inflation factor (VIF) analysis. All VIF values were within acceptable limits, indicating no significant collinearity concerns. In addition, the final model demonstrated a good model fit, with −2 Log-Likelihood (−2LL) of 28.308 and a pseudo $R^2$ of 0.512, indicating strong explanatory power. The Hosmer-Lemeshow goodness-of-fit test yielded a $\chi^2$ (8)=4.356, p-value = 0.824, suggesting excellent calibration and no evidence of poor fit; [c] To indicate non-invasive or invasive MV at the referring hospital or during transportation; [d] To indicate family members, relatives, neighbors, layperson, police, or passers-by; [e] To indicate an alternative method for oxygen supplementation, for instance, employing a manual resuscitator bag with an artificial airway; Abbreviations: **AOR**, adjusted odds ratio; **CCI**, Charlson Comorbidity Index; **CI**, confidence interval; **ECAT**, extracorporeal cytokine adsorption therapy; **EMS**, emergency medical services; **MV**, mechanical ventilation; **NA**, not available; **OR**, odds ratio; **PaO$_2$/FiO$_2$**, the ratio of arterial oxygen partial pressure to fractional-inspired oxygen; **SOFA Score**, Sequential Organ Failure Assessment Score.

or noninvasive MV during transport, prior research in Vietnam identifies suboptimal pre- and inter-hospital care as a risk factor for poor outcomes [60]. Therefore, improving patient transfer services, including access to critical interventions (e.g., HFNC, intubation, MV) and optimized ambulance systems, is essential.

In our study, the rate of patients receiving invasive MV (Table 5) is comparable to the LUNG SAFE study (84.5%; 2377/2813) [9], indicating a similar reliance on invasive MV for managing ARDS. However, a higher proportion of patients in our cohort received NMBAs (Table 5) compared to LUNG SAFE (21.7%; 516/2377) [9]. This discrepancy may be attributed to the lower PaO$_2$/FiO$_2$ ratio observed in our cohort (Table 4), suggesting more severe hypoxemia and a perceived need for more profound sedation and paralysis to optimize ventilator synchrony. Despite their use, NMBAs are not routinely recommended for ARDS due to limited evidence supporting their benefits and potential risks [61], including HAP and sepsis [62,63], which were prevalent complications in our study (Table 6). These complications underscore the need for cautious use of NMBAs, with careful monitoring to mitigate associated risks. ECMO was utilized in a significant subset of our patients (Table 5), at a higher rate than reported in the LUNG SAFE study (3.2%; 76/2377) [9]. This discrepancy may reflect greater illness severity or greater ECMO availability in our setting. A retrospective analysis of the Extracorporeal Life Support Organization Registry (ELSO) indicates that hospitals with a higher volume of ECMO procedures tend to have lower mortality rates [56], likely due to accumulated clinical experience and refined protocols. Optimal ECMO outcomes depends on lung-protective ventilation, prone positioning, and vigilant management of complications [56,57]. Although detailed ECMO data was unavailable in this study, previous research in Vietnam reveals that patients often require prolonged support, which heightens the risk of bleeding [64], infection [65], and circuit-related complications [66]. These findings underscore the importance of specialized training and robust infrastructure. In resource-limited settings such as Vietnam, the implementation of ECMO is hindered by high costs, limited skilled personnel, and inadequate critical care capacity. Standardized protocols for patient selection and management, aligned with ELSO guidelines, are essential to improving outcomes. While integrating ECMO with evidence-based interventions is crucial [57], our study did not demonstrate a mortality benefit among ARDS patients receiving ECMO. This may be due to inconsistent use of complementary strategies or delays in initiating ECMO. To advance ARDS care in Vietnam, there is a pressing need for strengthened critical care systems, context-specific protocols, and further research into the cost-effective application of ECMO.

In patients with ARDS, mortality rates tend to increase as the severity of hypoxemia worsens [19,36,67,68]. Our univariable analysis revealed a significant association between the PaO$_2$/FiO$_2$ ratio and hospital mortality (Table 8). However, this association did not hold in the multivariable analysis (Tables 8 and S14 as shown in S1 File), indicating that the PaO$_2$/FiO$_2$ ratio has limited prognostic value. This may be attributed to its poor discriminatory performance for predicting hospital mortality, as shown by an AUROC of 0.592 (95% CI: 0.528–0.656), a cutoff point of ≥ 121.1 mmHg, a sensitivity of 75.7%, a specificity of 46.3%, and a p-value of 0.004 (S13 Table as shown in S1 File). These findings cast doubt on the reliability of the PaO$_2$/FiO$_2$ ratio as a prognostic marker [69]. Similar inconsistencies have been reported internationally;

for example, a study in China reported an AUROC of 0.865 [70], while an Italian study found a lower AUROC of 0.688 [71]. In contrast, a SOFA score of ≥ 9.5 at admission was independently associated with a higher risk of hospital mortality (S14 Table as shown in S1 File), reinforcing the notion that infection and MODS are more reliable predictors of outcomes than respiratory parameters alone [15,72–76]. ARDS is increasingly recognized as a syndrome of systemic immune dysregulation. Although the concept of a cytokine storm was initially proposed [77], later studies have shown that systemic cytokine levels in ARDS patients are generally lower than previously thought, except in severe COVID-19 cases [26,78]. The use of ECAT has been explored for immune modulation. In our study, detailed ECAT data were limited. Still, its application was associated with higher hospital mortality (S14 Table in S1 File), likely because it was used as rescue therapy in critically ill patients (S15 Table in S1 File), suggesting confounding by indication rather than direct harm. A systematic review of hemadsorption (a form of ECAT) in ARDS patients on veno-venous ECMO highlighted potential benefits, including reduced inflammation and improved oxygenation [79]. However, the evidence is confined to small cohorts with limited long-term mortality data. A meta-analysis further supported the role of adjunctive hemadsorption in improving clinical outcomes in ARDS, but called for larger trials [80]. In resource-limited settings, such as Vietnam, where ECAT is reserved for severe cases due to cost and availability constraints, LMIC-focused studies are needed to evaluate its cost-effectiveness and integration with standard care. Prone positioning, utilized in nearly a quarter of our cohort (Table 5), primarily for patients with severe hypoxemia (S16 Table in S1 File). This approach aligns with established guidelines for moderate-to-severe ARDS, as it improves ventilation-perfusion matching and reduces ventilator-induced lung injury [57]. A narrative review on biomarkers in COVID-19 and non-COVID-19 ARDS supports their role in reducing mortality in severe cases, particularly when used early and for prolonged periods [81]. A systematic review and meta-analysis demonstrated that combining prone positioning with ECMO improves short-term survival in severe ARDS, with greater benefits in non-COVID patients [82]. However, barriers in our setting, such as staffing shortages and pandemic-related infection control measures, may have limited wider use, underscoring the need for LMIC-specific training programs. ECMO was applied to a notable subset of our patients (Table 5), exceeding the rate in the LUNG SAFE study (3.2%; 76/2377) [9], possibly reflecting higher disease severity or better local access. A review of ECMO in severe ARDS emphasizes its role in stabilizing refractory hypoxemia [83]. Still, it stresses careful patient selection and multidisciplinary management to address amplified complications, such as bleeding and infection, in resource-limited environments. A review of hypoxemic respiratory failure in cardiac patients further highlights these challenges [84]. Although HFNC was not explicitly detailed in our dataset (potentially under oxygen supplements in Table 5), it is increasingly supported as a non-invasive option for mild-to-moderate ARDS to delay intubation. A multicenter prospective cohort study showed that early prone positioning with HFNC or noninvasive ventilation reduced intubation rates by up to 50% in patients with moderate-to-severe ARDS, improving oxygenation and comfort [85]. Recent studies reinforce the efficacy of prone positioning in awake HFNC patients for COVID-19 ARDS, reducing MV needs [86]. In Vietnam, amid ventilator shortages, HFNC could address early management gaps, but prospective LMIC trials are essential to assess feasibility and outcomes.

During the pandemic, the ARDS case-mix shifted significantly, with a marked decline in cases attributed to seasonal viruses, such as influenza A (H1N1) (Table 7). This decline was due to non-pharmaceutical interventions (NPIs) such as mask-wearing and social distancing [87], mirroring global trends [87,88], which contrasts with pre-pandemic data in Vietnam, where influenza was a primary cause of ARDS, underscoring the need for robust epidemiological surveillance to monitor such changes. Conversely, there was an influx of ARDS cases associated with SARS-CoV-2, particularly the Delta variant, as critically ill COVID-19 patients often faced delayed hospital admissions, leading to severe pneumonia, rapid ARDS progression, and high mortality [25,78]. This etiological shift resulted in a more homogeneous viral-driven ARDS cohort during the pandemic, which differed from the diverse pre-pandemic mix of bacterial and other viral pneumonias, and potentially influenced treatment protocols in resource-limited settings. There was also a significant reduction in the use of both noninvasive and invasive MV at referring hospitals (Table 7). This decline stemmed from resource constraints [29], including acute ventilator shortages in LMICs that restricted MV access, and infection

control measures to limit aerosol-generating procedures, aligning with reports of strained healthcare systems world-wide [78,88,89]. Such limitations exacerbated challenges in early respiratory support, prompting greater reliance on alternatives, such as HFNC, when available [90]. Transport practices evolved as reliance on ET use increased during inter-hospital transfers to tertiary centers (Table 7), reflecting adaptations to manage infectious risks, stabilize airways in deteriorating patients, and cope with overwhelmed local facilities amid surges. This trend, observed in other LMICs [91,92], often led to increased transfers for critical cases due to uneven resource distribution. Still, it also added logistical and psychological burdens on patients and providers [93,94]. Vietnam's shortages of ventilators and critical care infrastructure were key drivers of these changes. A notable shift in ARDS management was the reduced use of invasive MV on the first day of admission during the pandemic (Table 7). This change is consistent with evolving guidelines favoring noninvasive strategies, such as HFNC, for COVID-19-related ARDS to conserve ventilators and minimize intubation risks [57,90]. Pre-pandemic, invasive MV was standard [19,36]; this change arose from both resource constraints and emerging evidence [57], underscoring the need for context-specific protocols in low-resource settings. Despite these shifts, hospital mortality rates for ARDS remained stable between pre-pandemic and pandemic periods (Table 7), suggesting resilience in Vietnam's critical care system. However, rates were consistently high, exceeding the 35–45% reported in HICs [9], likely due to limited infrastructure, including shortages of trained intensivists, ventilators, and standardized protocols.

This study has several limitations that warrant careful consideration when interpreting the findings. *First*, the retrospective design limited data availability and completeness across key variables (S17 Table in S1 File). Transport-related information was available for only 214 patients and lacked critical details, such as transfer duration, en route care level, vehicle type, and interventions. These gaps introduce potential selection bias into the multivariable logistic regression analyses, as unmeasured confounders, such as transfer delays or variations in monitoring, could affect patients' arrival condition and hospital mortality, which undermines model robustness, potentially leading to over- or under-estimation of associations and unreliable inferences about transport's role in ARDS outcomes. Vietnam's pre- and inter-hospital transfer systems remain underdeveloped, lacking standardization, integration, and regulatory oversight [30,32,34,55,95], which hampers data collection for surveillance, quality improvement, and research and limits the generalizability of our findings to other LMICs with similar constraints. To interpret these transport limitations, we adapt Donabedian's healthcare quality framework, categorizing factors into structural, process, and outcome dimensions, thereby highlighting how deficiencies in LMIC inter-hospital transport for ARDS patients amplify risks, unlike in HICs with specialized retrieval teams and protocols. Structural quality (i) involves foundational resources, such as ambulance types (hospital-operated vs. private/informal), staffing (trained physicians vs. nurses/bystanders), and equipment (ventilators, oxygen, ECMO). Incomplete documentation in our study underscores LMIC challenges, such as resource scarcity, potentially underestimating risks from equipment failure or inadequate stabilization. Process quality (ii) covers protocols and actions, including pre-transfer stabilization, communication, and interventions like ET or MV. Our data showed limited ET use (60.1% of transported patients), possibly delaying respiratory support and worsening hypoxemia; non-standardized processes may bias toward poorer outcomes. Outcome quality (iii) includes transport events (e.g., hypoxia, hypotension) and metrics like hospital mortality. ET use during transport was independently associated with lower mortality (Tables 8 and S14 in S1 File), but missing details hinder causal inferences due to unadjusted confounders. Despite these issues, our findings suggest prioritizing ET could mitigate transfer risks. Future prospective studies should use standardized protocols, including mobile ECMO or ET/MV interventions, to optimize ARDS transport in LMICs. *Second*, this single-center study at a tertiary hospital in Hanoi involved a selected cohort of deteriorating ARDS patients referred from local facilities, potentially overestimating mortality. Excluding incomplete data compounded by selection bias limited applicability to broader Vietnamese or LMIC populations. *Third*, data limitations prevented assessment of the effects of specific treatments on mortality. However, this is the first study examining transport factors in adult ARDS in Vietnam, offering insights for clinicians and policymakers in similar settings. *Finally*, the sample size (353 patients, 21 covariates) met traditional events-per-variable thresholds of 10–20 [51,52]

but falls short of newer recommendations (e.g., 50, requiring at least 500 patients) [53,54]. Wide confidence intervals for ET's protective effect necessitate caution. Further studies with larger sample sizes and more diverse populations will be essential to validate and expand upon these findings, enhancing their reliability and relevance.

## Conclusion

This retrospective observational study examined a highly selected cohort of patients with ARDS presenting to a tertiary hospital in Hanoi, Vietnam, where mortality rates were high. Most patients were transferred from local hospitals; yet, critical transport interventions, such as ET and MV, were frequently absent. ET use during transport was independently associated with reduced mortality, whereas a SOFA score of ≥9.5 and ECAT during the ICU stay were predictors of increased risk. Improving outcomes will require strengthening pre- and inter-hospital care, enhancing ambulance services, ensuring timely respiratory support, and expanding access to critical care. Optimizing the management of ARDS, its complications, and underlying conditions across all levels of the healthcare system remains essential.

## Supporting information

**S1 File. Supplementary results.**
(PDF)

## Acknowledgments

We express our deepest gratitude to the staff of the Center for Emergency Medicine and the Center for Critical Care Medicine at Bach Mai Hospital for their invaluable assistance and support throughout the completion of this study. We also extend our sincere appreciation to the staff of the Faculty of Public Health at Thai Binh University of Medicine and Pharmacy for their statistical contributions and constructive advice. Finally, we would like to thank Miss Isis Fenner from the Hotchkiss School, Lakeville, Connecticut, USA, and Miss Hoang Kieu Anh Le from the Hanoi – Amsterdam High School for the Gifted, Hanoi, Vietnam, for their support in preparing our manuscript.

## Author contributions

**Conceptualization:** Co Xuan Dao, Chinh Quoc Luong, Son Ngoc Do.

**Data curation:** Chinh Quoc Luong, Quynh Thi Pham, Tai Thien Vu, Hau Thi Truong, Dai Quoc Khuong, Hien Duy Dang.

**Formal analysis:** Chinh Quoc Luong, Toshie Manabe, My Ha Nguyen, Dung Thi Pham, Nga Thu Phan, Loc The Vu.

**Investigation:** Co Xuan Dao, Chinh Quoc Luong, Toshie Manabe, My Ha Nguyen, Dung Thi Pham, Quynh Thi Pham, Tai Thien Vu, Hau Thi Truong, Dai Quoc Khuong, Hien Duy Dang, Tuan Anh Nguyen, Thach The Pham, Giang Thi Huong Bui, Cuong Van Bui, Quan Huu Nguyen, Thong Huu Tran, Tan Cong Nguyen, Khoi Hong Vo, Lan Tuong Vu, Nga Thu Phan, Loc The Vu, Cuong Duy Nguyen, Thom Thi Vu, Anh Dat Nguyen, Chi Van Nguyen, Tuan Quoc Dang, Binh Gia Nguyen, Son Ngoc Do.

**Methodology:** Co Xuan Dao, Chinh Quoc Luong, Toshie Manabe, My Ha Nguyen, Dung Thi Pham, Son Ngoc Do.

**Project administration:** Chinh Quoc Luong.

**Supervision:** Co Xuan Dao, Chinh Quoc Luong, Son Ngoc Do.

**Writing – original draft:** Chinh Quoc Luong.

**Writing – review & editing:** Co Xuan Dao, Chinh Quoc Luong, Toshie Manabe, Tuan Anh Nguyen, Thach The Pham, Giang Thi Huong Bui, Cuong Van Bui, Quan Huu Nguyen, Thong Huu Tran, Tan Cong Nguyen, Khoi Hong Vo, Lan Tuong Vu, Cuong Duy Nguyen, Thom Thi Vu, Anh Dat Nguyen, Chi Van Nguyen, Tuan Quoc Dang, Binh Gia Nguyen, Son Ngoc Do.

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
