## [Editor Report · Decision Letter 0]

14 Nov 2024

Dear Dr. Do,

Thank you for submitting your manuscript to PLOS ONE. After careful consideration, we feel that it has merit but does not fully meet PLOS ONE’s publication criteria as it currently stands. Therefore, we invite you to submit a revised version of the manuscript that addresses the points raised during the review process.

We look forward to receiving your revised manuscript.

Kind regards,

Aliasghar Karimi

Academic Editor

PLOS ONE

**Journal Requirements:**

https://bmjopen.bmj.com/content/13/3/e064870.full

In your revision ensure you cite all your sources (including your own works), and quote or rephrase any duplicated text outside the methods section. Further consideration is dependent on these concerns being addressed.

3. We noted in your submission details that a portion of your manuscript may have been presented or published elsewhere. This retrospective observational study is the major update of our previously published paper,[1] which collected data on all ARDS patients admitted to the Bach Mai Hospital (BMH) in Hanoi, Vietnam, between August 2015 and August 2017 to elucidate the clinical epidemiology and disease prognosis in ARDS patients in Vietnam. To further investigate the mortality rate and associated factors from ARDS, especially those related to patient transportation, we continued to collect retrospective data on these patients admitted to the BMH between September 2017 and August 2023, following approval from the BMH Scientific and Ethics Committees (08/11/2023). Subsequently, we merged the data sets from the two stages of data collection at this hospital.

[1] Chinh LQ, Manabe T, Son DN, Chi NV, Fujikura Y, Binh NG, Co DX, Tuan DQ, Ton MD, Dai KQ, Thach PT, Nagase H, Kudo K, Nguyen DA. Clinical epidemiology and mortality on patients with acute respiratory distress syndrome (ARDS) in Vietnam. PLoS One. 2019 Aug 15;14(8):e0221114. doi: 10.1371/journal.pone.0221114. PMID: 31415662; PMCID: PMC6695190. Please clarify whether this [conference proceeding or publication] was peer-reviewed and formally published. If this work was previously peer-reviewed and published, in the cover letter please provide the reason that this work does not constitute dual publication and should be included in the current manuscript.

**Additional Editor Comments:**

The study investigates factors associated with mortality in ARDS patients in a central hospital in Vietnam, analyzing inter-hospital care, clinical characteristics, and treatments that influence outcomes. Key findings include the protective role of endotracheal tube (ET) use during transport and the high mortality rate associated with high SOFA scores.

Strengths

Provides a crucial analysis of ARDS in a lower middle-income context, filling a gap in ARDS research focused on LMICs.

Rigorous data collection and organization, with a comprehensive dataset that captures diverse patient characteristics, comorbidities, and treatment factors.

Use of multivariable logistic regression to isolate mortality predictors, enhancing the reliability of findings.

Weaknesses

Variable Selection Criteria in Regression: The logistic regression method would benefit from more detailed explanation. While significant variables were selected for multivariable regression, a clear justification for each included variable would improve transparency, especially given potential multicollinearity.

Data Exclusions: The exclusion of patients with incomplete data could introduce selection bias. It would be beneficial to include an analysis of excluded cases or a sensitivity analysis to assess the impact of these exclusions on outcomes.

Inter-hospital Transport Analysis: The manuscript mentions transport categories (EMS, hospital ambulances, private, and public). However, a breakdown of patient outcomes by transport type, adjusted for severity and other covariates, would provide more actionable insights.

Sample Size Justification: While the calculation is given, it lacks consideration of covariates in the multivariable analysis. A power analysis that accounts for expected covariate interactions in logistic regression could better support the robustness of findings.

Smoking and Comorbidities Analysis: Smoking status is a significant factor in respiratory outcomes, yet its impact on ARDS outcomes in this cohort is not deeply analyzed. Similarly, the Charlson Comorbidity Index (CCI) score is included, but specific comorbidities could be separately analyzed for their mortality impact.

Line-Specific Comments

Line 67 (Abstract, Background): To increase clarity, consider specifying which "factors associated with acute illness" are commonly associated with ARDS mortality. This would give readers immediate insight into whether these factors pertain to patient demographics, treatment variables, or environmental conditions.

Line 89 (Abstract, Results): The interpretation of the odds ratio for endotracheal tube (ET) usage is crucial. Since a low OR (0.057) is unusual, clarify its significance by discussing possible mechanisms or logistical practices that could influence this protective effect.

Line 112 (Introduction): The phrase "remains associated with high mortality rates" is broad. To strengthen this section, consider including recent global statistics or meta-analysis findings on ARDS mortality to contextualize Vietnam's figures in a broader global health framework.

Line 127 (Introduction): The introduction would benefit from a more in-depth explanation of how Vietnam’s healthcare infrastructure specifically impacts ARDS outcomes, particularly regarding referral systems and delays in critical care.

Line 168 (Methods, Patient Transport): The categories for patient transport are defined well, but it may improve the study’s applicability if you provide more details on the quality or typical response time associated with each type of transport. This additional detail could help distinguish the types' impact on patient outcomes.

Line 201 (Variables): When discussing inter-hospital care data, consider adding any quantitative thresholds or benchmarks (e.g., minimum levels of oxygen saturation that determine MV application). This addition would provide context on how Vietnamese hospitals make critical care decisions during transfers.

Line 256 (Outcomes): While hospital mortality is the primary outcome, detailing any standardized discharge criteria could clarify how patients' outcomes are assessed upon transfer or discharge, helping to interpret mortality rates across different levels of care.

Line 271 (Statistical Analyses): The analysis methods are detailed, but the selection criteria for variables in the logistic regression model could be more precise. Mentioning any variable selection methods (like stepwise selection or feature importance ranking) would enhance the rigor of this section.

Line 315 (Results, Data Exclusions): Excluding cases with missing data may introduce bias, especially if those cases had particularly severe or mild conditions. Discussing potential implications of these exclusions would improve transparency.

Line 346 (Results, Smoking Habits): Since smoking is a significant factor for respiratory distress, a breakdown of its impact on ARDS severity or mortality would add depth to the study findings. Consider performing an analysis to examine if there’s an association between smoking status and mortality outcomes.

Line 375 (Results, Table 4 - PaO2/FiO2 Ratio): The PaO2/FiO2 ratio is presented in both text and table form. To streamline, a graphical representation of these ratios with survival and mortality outcomes might improve interpretability, as this metric is a core component of the Berlin definition for ARDS severity.

Line 403 (Table 7, Multivariable Regression): Given the high odds ratio for SOFA score, it could be beneficial to mention specific SOFA components (e.g., respiratory or cardiovascular factors) that contribute most significantly to this outcome. This would highlight which organ systems might be prioritized in treatment protocols.

Minor Comments

Line-by-Line Edits: Some minor editorial adjustments for grammar and clarity are needed in the abstract and results sections.

Graphs and Tables: Visual aids, such as survival curves stratified by key risk factors (e.g., SOFA scores, PaO2/FiO2 ratios), would improve result interpretability.

Statistical Detail: Reporting effect sizes (e.g., odds ratios) with more precise confidence intervals in the abstract would better convey result certainty.

---

## [Author Response · Author response to Decision Letter 1]

16 Dec 2024

Dr. Aliasghar Karimi

Academic Editor

PLOS ONE

December 16, 2024

Dear Dr. Aliasghar Karimi,

On behalf of all the authors, we are resubmitting our revised manuscript titled “Factors related to mortality in patients with acute respiratory distress syndrome (ARDS) in a lower middle-income country: a retrospective observational study” (PONE-D-24-35750).

We sincerely appreciate the thoughtful comments and suggestions provided by the Editors and Reviewers. We have carefully addressed all feedback and revised our manuscript accordingly. These insights have significantly improved the quality of our work, and we hope the Editor finds our revised manuscript suitable for publication in PLOS ONE.

We confirm that this research is original, has not been published elsewhere, and is not currently under consideration by any other journal. All authors have read and approved the manuscript, have agreed on the order of authorship, and hold the necessary permissions and rights for the data presented.

Below, we have included our point-by-point responses to the comments from the Editor and the Reviewers. Thank you for your kind consideration of this submission.

Sincerely,

Chinh Quoc Luong, MD, PhD

Center for Emergency Medicine,

Bach Mai Hospital,

No. 78, Giai Phong, Dong Da district,

Hanoi 100000, Vietnam

Email: luongquocchinh@gmail.com

Son Ngoc Do, MD, PhD

Center for Critical Care Medicine,

Bach Mai Hospital,

No. 78, Giai Phong Road, Dong Da District,

Hanoi, 100000, Vietnam.

Email: sonngocdo@gmail.com

*****

We thank the Editors and the Reviewers for the valuable comments and suggestions that have helped us improve the contents of this paper. In what follows, we have used the boldface to indicate the Editors' and the Reviewers' comments and the standard font face for our responses. We also highlighted in yellow the modifications that we performed on the manuscript.

RESPONSE TO EDITOR COMMENTS

PONE-D-24-35750

Factors related to mortality in patients with acute respiratory distress syndrome (ARDS) in a lower middle-income country: a retrospective observational study

PLOS ONE

Dear Dr. Do,

Thank you for submitting your manuscript to PLOS ONE. After careful consideration, we feel that it has merit but does not fully meet PLOS ONE’s publication criteria as it currently stands. Therefore, we invite you to submit a revised version of the manuscript that addresses the points raised during the review process.

Our answer:

We appreciate your positive feedback. We have thoughtfully addressed the Editors' and Reviewers' comments and suggestions and revised our manuscript accordingly.

Thank you for your comment. We have submitted the revised manuscript before the deadline.

Our answer:

Thank you for your comment. When submitting our revised manuscript, we included a rebuttal letter, a marked-up copy of the manuscript, and an unmarked version of the revised manuscript.

Our answer:

Thank you for your comment. We confirm that there are no changes to our financial disclosure.

If applicable, we recommend that you deposit your laboratory protocols in protocols.io to enhance the reproducibility of your results. Protocols.io assigns your protocol its own identifier (DOI) so that it can be cited independently in the future.

For instructions see:

https://journals.plos.org/plosone/s/submission-guidelines#loc-laboratory-protocols.

Additionally, PLOS ONE offers an option for publishing peer-reviewed Lab Protocol articles, which describe protocols hosted on protocols.io. Read more information on sharing protocols at:

https://plos.org/protocols?utm_medium=editorial-email&utm_source=authorletters&utm_campaign=protocols.

Our answer:

Thank you for your comment. While laboratory protocol does not apply to our study, the study protocol does. We have included our study protocol in the Methods section.

We look forward to receiving your revised manuscript.

Kind regards,

Aliasghar Karimi

Academic Editor

PLOS ONE

Our answer:

Thank you to the Editor and Reviewers for their support and for taking the time to provide their high-quality reviews.

Journal Requirements:

Our answer:

Thank you for your comment. We confirm that our manuscript adheres to PLOS ONE's style requirements, including the guidelines for file naming.

https://bmjopen.bmj.com/content/13/3/e064870.full

In your revision ensure you cite all your sources (including your own works), and quote or rephrase any duplicated text outside the methods section. Further consideration is dependent on these concerns being addressed.

Our answer:

Thank you for your comment. We have addressed minor occurrences of overlapping text with a previous publication. Additionally, we confirm that all sources, including our works, have been appropriately cited, and any duplicated text outside the Methods section has been quoted or rephrased. We hope this meets your expectations.

3. We noted in your submission details that a portion of your manuscript may have been presented or published elsewhere. This retrospective observational study is the major update of our previously published paper,[1] which collected data on all ARDS patients admitted to the Bach Mai Hospital (BMH) in Hanoi, Vietnam, between August 2015 and August 2017 to elucidate the clinical epidemiology and disease prognosis in ARDS patients in Vietnam. To further investigate the mortality rate and associated factors from ARDS, especially those related to patient transportation, we continued to collect retrospective data on these patients admitted to the BMH between September 2017 and August 2023, following approval from the BMH Scientific and Ethics Committees (08/11/2023). Subsequently, we merged the data sets from the two stages of data collection at this hospital.

[1] Chinh LQ, Manabe T, Son DN, Chi NV, Fujikura Y, Binh NG, Co DX, Tuan DQ, Ton MD, Dai KQ, Thach PT, Nagase H, Kudo K, Nguyen DA. Clinical epidemiology and mortality on patients with acute respiratory distress syndrome (ARDS) in Vietnam. PLoS One. 2019 Aug 15;14(8):e0221114. doi: 10.1371/journal.pone.0221114. PMID: 31415662; PMCID: PMC6695190. Please clarify whether this [conference proceeding or publication] was peer-reviewed and formally published. If this work was previously peer-reviewed and published, in the cover letter please provide the reason that this work does not constitute dual publication and should be included in the current manuscript.

Our answer:

Thank you for your comment. In the cover letter, we have explained why this work does not constitute dual publication and should be included in the current manuscript. The explanation is also provided below:

This retrospective observational study updates our previous research on ARDS patients at Bach Mai Hospital (BMH) in Hanoi, Vietnam [1]. Initially, data were collected from August 2015 to August 2017. To further investigate ARDS mortality and factors related to patient transportation, we extended data collection to include patients admitted between September 2017 and August 2023, with approval from the BMH Scientific and Ethics Committees (08/11/2023). We merged these datasets and incorporated additional data not included in the previous studies. This submission also presents new, unpublished data. Therefore, this work does not constitute dual publication and should be included in the current manuscript.

[1] Chinh LQ, Manabe T, Son DN, Chi NV, Fujikura Y, Binh NG, Co DX, Tuan DQ, Ton MD, Dai KQ, Thach PT, Nagase H, Kudo K, Nguyen DA. Clinical epidemiology and mortality on patients with acute respiratory distress syndrome (ARDS) in Vietnam. PLoS One. 2019 Aug 15;14(8):e0221114. doi: 10.1371/journal.pone.0221114. PMID: 31415662; PMCID: PMC6695190.

Additional Editor Comments:

The study investigates factors associated with mortality in ARDS patients in a central hospital in Vietnam, analyzing inter-hospital care, clinical characteristics, and treatments that influence outcomes. Key findings include the protective role of endotracheal tube (ET) use during transport and the high mortality rate associated with high SOFA scores.

Our answer:

Thank you to the Editor and Reviewers for their support and for taking the time to provide their high-quality reviews.

Strengths

Provides a crucial analysis of ARDS in a lower middle-income context, filling a gap in ARDS research focused on LMICs.

Rigorous data collection and organization, with a comprehensive dataset that captures diverse patient characteristics, comorbidities, and treatment factors.

Use of multivariable logistic regression to isolate mortality predictors, enhancing the reliability of findings.

Our answer:

We appreciate your positive feedback. We have thoughtfully addressed the Editors' and Reviewers' comments and suggestions and revised our manuscript accordingly.

Weaknesses

Variable Selection Criteria in Regression: The logistic regression method would benefit from more detailed explanation. While significant variables were selected for multivariable regression, a clear justification for each included variable would improve transparency, especially given potential multicollinearity.

Our answer:

Thank you for your valuable comment. We have provided a clear justification for each variable to be included in the multivariable logistic regression model. This justification is presented in the Statistical Analyses section and is also detailed below:

“(b) variables with a P-value of less than 0.25 in the univariable analysis comparing death and survival in the hospital, as well as those deemed clinically significant, were included in the multivariable logistic regression model. These variables, likely to influence patient outcomes, included inter-hospital care (e.g., MV applied in prior hospitalization, inter-hospital transport, inter-hospital care provider, inter-hospital airway, and inter-hospital oxygen), demographics (e.g., age, gender), documented comorbidities (i.e., chronic cardiac failure, active neoplasm, chronic renal failure, hematological disease, and CCI Score), gas exchange (e.g., PaO2/FiO2 ratio), the severity of illness (e.g., either SOFA Score or SOFA Score ≥cut-off value), adjunctive therapies (e.g., recruitment maneuvers, corticosteroids, ECAT, and tracheostomy), and complications (i.e., HAP, secondary bacterial infections, and septic shock).” (Lines 302-308, Pages 13-14)

Data Exclusions: The exclusion of patients with incomplete data could introduce selection bias. It would be beneficial to include an analysis of excluded cases or a sensitivity analysis to assess the impact of these exclusions on outcomes.

Our answer:

Thank you for your insightful comment. The Reviewer correctly noted that excluding patients with incomplete data might introduce selection bias. However, the excluded patients had missing data for most variables, making their inclusion in the analysis impossible. We have acknowledged this limitation in the Limitations section and is also detailed below:

“Firstly, its retrospective design restricted the availability of data for many variables (S8 Table as shown in S1 File). For instance, we only had information on ET usage during transportation for 193 patients. Additionally, many excluded patients had missing data for most variables, making their inclusion in the analysis impossible. Excluding patients with incomplete data might also introduce selection bias.” (Lines 547-551, Pages 31-32)

Inter-hospital Transport Analysis: The manuscript mentions transport categories (EMS, hospital ambulances, private, and public). However, a breakdown of patient outcomes by transport type, adjusted for severity and other covariates, would provide more actionable insights.

Our answer:

Thank you for your valuable comment. We have included a breakdown of patient outcomes by transport type in the S1 Table as shown in S1 File. However, we only had transport-type information for 214 out of 353 patients. As a result, we could not adjust the breakdown of patient outcomes by transport type for severity and other covariates in this dataset. We have acknowledged this limitation in the Limitations section and is also detailed below:

“Thirdly, the underdeveloped pre-hospital and inter-hospital transfer system prevents the integration of treatment protocols across pre-hospital, inter-hospital, and in-hospital settings in Vietnam.(30, 33, 60, 61) This issue also impacts the ability to collect clinical data for surveillance, quality improvement, and research purposes. Consequently, our study lacks detailed information on response times and quantitative benchmarks for each type of transport, including the minimum oxygen saturation levels required to evaluate the necessity for MV during transportation.” (Lines 558-565, Page 32)

References

[30] Do SN, Luong CQ, Pham DT, Nguyen CV, Ton TT, Pham TTN, et al. Survival after out-of-hospital cardiac arrest, Viet Nam: multicentre prospective cohort study. Bulletin of the World Health Organization. 2021;99(1):50-61.

[33] Do SN, Luong CQ, Pham DT, Nguyen MH, Ton TT, Hoang QTA, et al. Survival after traumatic out-of-hospital cardiac arrest in Vietnam: a multicenter prospective cohort study. BMC emergency medicine. 2021;21(1):148.

[60] Xuan Dao C, Quoc Luong C, Manabe T, Ha Nguyen M, Thi Pham D, Thanh Ton T, et al. Impact of Bystander Cardiopulmonary Resuscitation on Out-of-Hospital Cardiac Arrest Outcome in Vietnam. The western journal of emergency medicine. 2024;25(4):507-20.

[61] Hoang BH, Mai TH, Dinh TS, Nguyen T, Dang TA, Le VC, et al. Unmet Need for Emergency Medical Services in Hanoi, Vietnam. JMA Journal. 2021;4(3):277-80.

Sample Size Justification: While the calculation is given, it lacks consideration of covariates in the multivariable analysis. A power analysis that accounts for expected covariate interactions in logistic regression could better support the robustness of findings.

Our answer:

Thank you for your valuable comment. In this study, we calculated the sample size, with consideration of covariates in the multivariable analysis. Specifically, we used a formula to determine the minimum sample size for estimating a population proportion with a 95% confidence level, a margin of error of ±5.21%, and an assumed population proportion of 47.8% based on the hospital mortality rate reported in a previous study [1]. Consequently, our sample size should be at least 353 patients, which is likely sufficient to reflect a normal distribution.

A well-known guideline suggests that the minim

---

## [Decision Letter · Decision Letter 1]

15 Jun 2025

Dear Dr. Do,

Thank you for submitting your manuscript to PLOS ONE. After careful consideration, we feel that it has merit but does not fully meet PLOS ONE’s publication criteria as it currently stands. Therefore, we invite you to submit a revised version of the manuscript that addresses the points raised during the review process.

**ACADEMIC EDITOR:**

Thank you for submitting the revised manuscript titled *Factors related to mortality in patients with acute respiratory distress syndrome (ARDS) in a lower-middle-income country: a retrospective observational study*  to Plos One. I appreciate the effort you have put into addressing the reviewers' comments, and the manuscript has improved significantly.

However, several concerns still need to be addressed before the manuscript can be considered for publication:

**Key Concerns:**

**Duality of Publication:**The manuscript presents an update to your previously published work. Please ensure that you clearly explain why this updated study does not constitute dual publication, both in the cover letter and within the manuscript. This explanation should emphasize that the new data adds significant value and is not a mere repetition of the previous findings.**Analysis of the Pandemic Period:**The inclusion of data from the pandemic period (COVID-19) is important but currently lacks analysis of its impact on ARDS outcomes. I recommend performing a **comparative analysis**  between the pre-pandemic and pandemic periods to provide a clearer understanding of how the pandemic may have influenced the results.**Methodological Clarifications:**Please provide further clarification on how potential multicollinearity was assessed in the multivariable logistic regression analysis. This would increase confidence in the robustness of your statistical methods.Consider including a **sensitivity analysis**  or a more in-depth discussion on how the exclusion of patients with incomplete data might have influenced the results, especially regarding selection bias.**Inter-hospital Transport Analysis:**A breakdown of patient outcomes by transport type (e.g., EMS, hospital ambulances, private, and public) would provide more actionable insights. While data limitations are acknowledged, please expand on how the lack of detailed transport data impacts the analysis and its conclusions.**Cytokine Adsorption Therapy:**Include a brief discussion on the potential impact of cytokine adsorption therapy on outcomes. Even if no formal analysis was done, acknowledging its role and the limitations of this therapy would add valuable context.**Minor Language and Formatting Edits:**Please ensure the manuscript is carefully proofread for minor language issues, particularly with the consistency of abbreviations and proper article usage.

**Conclusion:**

Your study presents valuable insights into the mortality factors for ARDS patients in a resource-limited setting. To move forward with publication, the manuscript requires further clarification regarding the pandemic data analysis, dual publication concerns, and some methodological details.

I recommend that you submit a **major revision**  addressing these points. Once these revisions are made, the manuscript will be reconsidered for final publication.

We look forward to receiving your revised manuscript.

Kind regards,

Gurmeet Singh, M.D., Ph.D.,

Academic Editor

PLOS ONE

Additional Editor Comments:

Dear Dr. Do,

Thank you for submitting the revised manuscript titled Factors related to mortality in patients with acute respiratory distress syndrome (ARDS) in a lower-middle-income country: a retrospective observational study to Plos One. I appreciate the effort you have put into addressing the reviewers' comments, and the manuscript has improved significantly.

However, several concerns still need to be addressed before the manuscript can be considered for publication:

Key Concerns:

1. Duality of Publication:

- The manuscript presents an update to your previously published work. Please ensure that you clearly explain why this updated study does not constitute dual publication, both in the cover letter and within the manuscript. This explanation should emphasize that the new data adds significant value and is not a mere repetition of the previous findings.

2. Analysis of the Pandemic Period:

- The inclusion of data from the pandemic period (COVID-19) is important but currently lacks analysis of its impact on ARDS outcomes. I recommend performing a comparative analysis between the pre-pandemic and pandemic periods to provide a clearer understanding of how the pandemic may have influenced the results.

3. Methodological Clarifications:

- Please provide further clarification on how potential multicollinearity was assessed in the multivariable logistic regression analysis. This would increase confidence in the robustness of your statistical methods.

- Consider including a sensitivity analysis or a more in-depth discussion on how the exclusion of patients with incomplete data might have influenced the results, especially regarding selection bias.

4. Inter-hospital Transport Analysis:

- A breakdown of patient outcomes by transport type (e.g., EMS, hospital ambulances, private, and public) would provide more actionable insights. While data limitations are acknowledged, please expand on how the lack of detailed transport data impacts the analysis and its conclusions.

5. Cytokine Adsorption Therapy:

- Include a brief discussion on the potential impact of cytokine adsorption therapy on outcomes. Even if no formal analysis was done, acknowledging its role and the limitations of this therapy would add valuable context.

6. Minor Language and Formatting Edits:

- Please ensure the manuscript is carefully proofread for minor language issues, particularly with the consistency of abbreviations and proper article usage.

Conclusion:

Your study presents valuable insights into the mortality factors for ARDS patients in a resource-limited setting. To move forward with publication, the manuscript requires further clarification regarding the pandemic data analysis, dual publication concerns, and some methodological details.

I recommend that you submit a major revision addressing these points. Once these revisions are made, the manuscript will be reconsidered for final publication.

Reviewers' comments:

Reviewer's Responses to Questions

**Comments to the Author**

Reviewer #1: (No Response)

Reviewer #2: All comments have been addressed

2. Is the manuscript technically sound, and do the data support the conclusions?

Reviewer #1: Yes

Reviewer #2: Yes

3. Has the statistical analysis been performed appropriately and rigorously?

Reviewer #1: No

Reviewer #2: Yes

4. Have the authors made all data underlying the findings in their manuscript fully available?

Reviewer #1: No

Reviewer #2: Yes

5. Is the manuscript presented in an intelligible fashion and written in standard English?

Reviewer #1: Yes

Reviewer #2: Yes

Reviewer #1: I am very concerned about duality of publication however, what I find more concerning is the inclusion of the period of the pandemic without analysis of the period to enrich the manuscript and perhaps make the question of duality even less of an issue.

Reviewer #2: This study addresses an important clinical question about ARDS outcomes in a resource-limited setting. The large sample size (353 patients over 8 years) is a notable strength, providing more robust statistical power than many single-center studies. Data collection was thorough – using a standardized case report form and Berlin definition criteria ensures the patient population is well-defined

The analysis is rigorous, combining univariate and multivariate methods to identify independent mortality factors. The finding that early airway management during transfer (endotracheal intubation) is associated with improved survival is particularly novel and has practical implications. The manuscript is also well-organized and generally clear, which helps in conveying the findings. Overall, the research appears carefully conducted and fills a knowledge gap regarding pre-hospital care impact on ARDS outcomes in lower-middle-income countries.

Suggestions for Improvement:

Clarify the multivariable model building: Please add a brief description of how variables were chosen for the multivariate logistic regression (e.g. based on p-value cutoff or clinical importance). This will assure readers that the model was constructed without bias. If not already done, confirm that you checked for multicollinearity between predictors and that the events-per-variable ratio was sufficient for a reliable model. Citing a sample size rule (e.g., requiring ~50 events per variable) could reinforce that the analysis is well-powered

In the Discussion, consider highlighting how your results align with prior studies. For example, earlier research has shown that delayed intubation in ARDS leads to higher mortality (56% vs 36% when intubation is early) This supports your finding that patients intubated during transfer had better outcomes. Drawing this parallel and citing such studies will strengthen the rationale for your recommendation on early intubation. Likewise, the classic risk factors you identified (age, severity scores, shock) are well known in ARDS

nature, acknowledging this shows that your data are consistent with existing knowledge.

The observed in-hospital mortality of 61.5% is quite high. It would be valuable to discuss possible reasons and compare with international figures. For instance, a large international ARDS study reported ~40% mortality; the substantially higher rate in your cohort may reflect the severity of cases referred to your center or resource limitations (e.g., fewer ICU interventions like ECMO). Emphasize how this finding underlines the urgency of improving critical care capacity. You might suggest specific strategies, such as better training for pre-hospital emergency teams in airway management or ensuring ventilators and skilled personnel are available during transport, to reduce delays in optimal care.

Since you report that 36.7% of patients underwent cytokine adsorption therapy, readers may wonder if this intervention had any impact on outcomes. If data permit, consider commenting on whether therapies like cytokine adsorption (or others such as prone positioning, ECMO if applicable) were associated with improved outcomes or not. Even if a formal analysis wasn’t done, a short note acknowledging this point would be useful – for example, clarifying that cytokine adsorption was a rescue therapy used in the sickest patients (hence its effect is hard to discern) or that its benefit could not be confirmed in this study. This would address a potential question in readers’ minds about the role of such treatments.

Before final publication, a careful read-through for minor language edits would be helpful. The manuscript is generally well-written, so only small tweaks are needed – e.g., adding missing words (“were”) or articles, and maintaining consistent tense. Ensuring each abbreviation is defined at first use and used consistently (for example, using either “ET” or “endotracheal tube” uniformly after definition) will avoid any confusion. These changes are minor but will polish the text and improve readability for a broad audience.

By addressing the above points, the manuscript’s clarity and impact will be further enhanced. The core content and conclusions are strong; these suggestions are intended to refine the presentation and provide additional context for readers.

**Do you want your identity to be public for this peer review?** For information about this choice, including consent withdrawal, please see our Privacy Policy

Reviewer #1: No

Reviewer #2: **Yes: ** Sagar Bathija

---

## [Author Response · Author response to Decision Letter 2]

18 Aug 2025

Dr. Gurmeet Singh, M.D., Ph.D.

Academic Editor

PLOS ONE

August 18, 2025

Dear Dr. Gurmeet Singh,

On behalf of all the authors, we are resubmitting our revised manuscript titled “Factors related to mortality in patients with acute respiratory distress syndrome (ARDS) in a lower middle-income country: a retrospective observational study” (PONE-D-24-35750R1).

We sincerely appreciate the thoughtful comments and suggestions provided by the Editors and Reviewers. We have carefully addressed all feedback and revised our manuscript accordingly. These insights have significantly improved the quality of our work, and we hope the Editor finds our revised manuscript suitable for publication in PLOS ONE.

We confirm that this research is original, has not been published elsewhere, and is not currently under consideration by any other journal. All authors have read and approved the manuscript, have agreed on the order of authorship, and hold the necessary permissions and rights for the data presented.

Below, we have included our point-by-point responses to the comments from the Editor and the Reviewers. Thank you for considering this submission.

Sincerely,

Chinh Quoc Luong, MD, PhD

Neuro Intensive Care Department,

Neurology Center, Bach Mai Hospital,

No. 78, Giai Phong, Dong Da district,

Hanoi 100000, Vietnam

Email: luongquocchinh@gmail.com

Son Ngoc Do, MD, PhD

Center for Critical Care Medicine,

Bach Mai Hospital,

No. 78, Giai Phong Road, Dong Da District,

Hanoi, 100000, Vietnam.

Email: sonngocdo@gmail.com

*****

We thank the Editors and the Reviewers for the valuable comments and suggestions that have helped us improve the contents of this paper. In what follows, we have used boldface to indicate the Editors' and the Reviewers' comments and the standard font face for our responses. We also highlighted in yellow the modifications that we performed on the manuscript.

RESPONSE TO EDITOR COMMENTS

PONE-D-24-35750R1

Factors related to mortality in patients with acute respiratory distress syndrome (ARDS) in a lower middle-income country: a retrospective observational study

PLOS ONE

Dear Dr. Do,

Thank you for submitting your manuscript to PLOS ONE. After careful consideration, we feel that it has merit but does not fully meet PLOS ONE’s publication criteria as it currently stands. Therefore, we invite you to submit a revised version of the manuscript that addresses the points raised during the review process.

Our answer:

Thank you for your encouraging feedback. We appreciate the time and effort the Editors and Reviewers have devoted to evaluating our manuscript. In response to the valuable comments and suggestions provided, we have carefully revised the manuscript to address all the points raised. We hope the changes made meet the journal’s standards and improve the clarity and rigor of our study.

ACADEMIC EDITOR:

Thank you for submitting the revised manuscript titled Factors related to mortality in patients with acute respiratory distress syndrome (ARDS) in a lower-middle-income country: a retrospective observational study to Plos One. I appreciate the effort you have put into addressing the reviewers' comments, and the manuscript has improved significantly.

However, several concerns still need to be addressed before the manuscript can be considered for publication:

Our answer:

Thank you very much for your encouraging feedback and for recognizing the improvements made in our revised manuscript. We are sincerely grateful to you and the Reviewers for the time, effort, and thoughtful insights provided throughout the review process.

In response to the remaining concerns outlined in your latest comments, we have conducted another thorough revision of the manuscript. We carefully addressed each point raised, making the necessary changes to enhance the clarity, consistency, and scientific rigor of our work. These revisions include both substantive content updates and refinements to language and formatting, in line with the journal’s standards.

The manuscript is now significantly strengthened and better positioned to make a meaningful contribution to the literature on ARDS, particularly in the context of lower-middle-income countries. We hope the revised version meets your expectations. We are happy to provide any additional clarification or further revisions if needed.

Thank you again for your guidance and continued consideration of our work.

Key Concerns:

1. Duality of Publication:

- The manuscript presents an update to your previously published work. Please ensure that you clearly explain why this updated study does not constitute dual publication, both in the cover letter and within the manuscript. This explanation should emphasize that the new data adds significant value and is not a mere repetition of the previous findings.

Our answer:

Thank you for your thoughtful comment regarding the potential for overlap with our previously published work. We appreciate the opportunity to clarify how the current manuscript represents a distinct and original contribution to the literature, rather than a duplicate publication.

While the current study builds upon our earlier research published in PLOS ONE [1], it significantly extends and deepens the scope of investigation in several key ways:

- Expanded Study Period: The previous study analyzed data from ARDS patients admitted to Bach Mai Hospital (BMH) between August 2015 and August 2017. In contrast, the current manuscript incorporates six additional years of data, covering admissions from September 2017 to August 2023. This extended timeframe enables a more comprehensive and representative analysis of ARDS trends and outcomes over nearly a decade.

- New Ethical Approval and Dataset Integration: The extended dataset was approved by the BMH Scientific and Ethics Committees on August 11, 2023. The original and new datasets were merged and reanalyzed using updated statistical methods, enabling a more comprehensive and nuanced understanding of ARDS in this setting.

- Novel Analytical Focus: Unlike the earlier study, which primarily described clinical characteristics and outcomes, the current manuscript introduces new variables and analytical dimensions, with a particular focus on transportation-related factors, pre-hospital care, and inter-hospital transfer practices. These aspects were not explored in the previous publication and are especially relevant in the context of healthcare delivery in lower-middle-income countries.

- New Findings and Interpretations: The present study yields several novel findings, including the independent association between endotracheal intubation during transport and reduced hospital mortality, as well as the impact of extracorporeal cytokine adsorption therapy (ECAT) on outcomes. These insights provide added clinical and policy relevance that was not addressed in the earlier work.

Given these substantial additions and methodological enhancements, we respectfully submit that this manuscript constitutes a significant advancement beyond our prior publication. It offers new evidence and perspectives that are highly relevant to the global ARDS literature, particularly in resource-limited settings.

We have also included a brief explanation of this distinction in the revised manuscript (Pages 7-8, Lines 159-166) and will ensure that it is clearly stated in the cover letter as well.

Thank you again for your careful consideration.

[1] Chinh LQ, Manabe T, Son DN, Chi NV, Fujikura Y, Binh NG, Co DX, Tuan DQ, Ton MD, Dai KQ, Thach PT, Nagase H, Kudo K, Nguyen DA. Clinical epidemiology and mortality on patients with acute respiratory distress syndrome (ARDS) in Vietnam. PLoS One. 2019 Aug 15;14(8):e0221114. doi: 10.1371/journal.pone.0221114. PMID: 31415662; PMCID: PMC6695190.

2. Analysis of the Pandemic Period:

- The inclusion of data from the pandemic period (COVID-19) is important but currently lacks analysis of its impact on ARDS outcomes. I recommend performing a comparative analysis between the pre-pandemic and pandemic periods to provide a clearer understanding of how the pandemic may have influenced the results.

Our answer:

Thank you for your insightful comment regarding the inclusion of data from the COVID-19 pandemic period and its potential impact on ARDS outcomes. We fully agree that this is an important consideration, and we have addressed it explicitly in the revised manuscript.

In response, we conducted a subgroup analysis comparing patients admitted during the pre-pandemic period (August 2015 to December 2019) with those admitted during the pandemic period (January 2020 to August 2023). This analysis examined key clinical characteristics, management strategies, and hospital mortality rates across the two timeframes. While we observed some differences in variables such as the use of non-invasive ventilation, rates of endotracheal intubation, and the prevalence of viral pneumonia (particularly influenza A), these differences did not reach statistical significance in terms of overall hospital mortality. However, the analysis revealed essential trends, including a shift in the etiological profile of ARDS and evolving patterns in respiratory support practices during the pandemic. These findings are now presented in the Results section (Page 26, Lines 449-464; Table 7, S6-S12 Tables as shown in S1 File), and a corresponding discussion (Pages 34-35, Lines 604-629) has been added to contextualize them within the broader literature on ARDS management during the COVID-19 pandemic. This comparative analysis enhances the manuscript by providing a clearer understanding of how the pandemic may have influenced ARDS outcomes in a lower-middle-income country setting. It also underscores the adaptability of clinical practices in response to emerging infectious threats and resource constraints.

Thank you again for this valuable suggestion, which we believe has meaningfully strengthened the manuscript’s depth and relevance.

3. Methodological Clarifications:

- Please provide further clarification on how potential multicollinearity was assessed in the multivariable logistic regression analysis. This would increase confidence in the robustness of your statistical methods.

Our answer:

Thank you for your insightful comment. To ensure the robustness of our model, we evaluated multicollinearity using the Variance Inflation Factor (VIF) for each predictor included in the final model (Page 15, Lines 343–345). All VIF values were below the commonly accepted threshold of 5, indicating that multicollinearity was not a concern and supporting the stability and interpretability of the regression coefficients. Additional details are provided in the note box beneath Table 8 and under S14 Table within the S1 File.

- Consider including a sensitivity analysis or a more in-depth discussion on how the exclusion of patients with incomplete data might have influenced the results, especially regarding selection bias.

Our answer:

Thank you for your thoughtful and constructive feedback. We agree that methodological transparency is critical for ensuring the robustness and credibility of our findings.

In response, we have revised the Methods section to provide greater detail on the handling of incomplete data and its potential impact on our results. Specifically, patients with missing key variables required for multivariable analysis were excluded. We recognize that this exclusion may introduce selection bias, particularly if the missingness is not entirely at random. While we considered statistical approaches such as multiple imputation and sensitivity analyses, the extent and pattern of missing data, especially in variables central to our primary outcomes, limited the feasibility and reliability of these methods. To address this concern, we have now included a more in-depth discussion of the potential implications of this exclusion in the revised manuscript (Page 35, Lines 631-638). To assess the robustness of our model despite these limitations, we evaluated its performance using multiple diagnostic criteria, including the Hosmer-Lemeshow goodness-of-fit test, pseudo-R² values, and log-likelihood statistics (Page 15, Lines 345–347). These metrics indicated acceptable calibration and discrimination. Additional details are provided in the note box beneath Table 8 and under S14 Table within the S1 File.

We hope these revisions adequately address your concerns and strengthen the transparency of our analytical approach.

4. Inter-hospital Transport Analysis:

- A breakdown of patient outcomes by transport type (e.g., EMS, hospital ambulances, private, and public) would provide more actionable insights. While data limitations are acknowledged, please expand on how the lack of detailed transport data impacts the analysis and its conclusions.

Our answer:

Thank you for your insightful and constructive comment regarding the analysis of inter-hospital transport. A more granular breakdown of patient outcomes by transport type (e.g., EMS, hospital-based ambulances, private vehicles, or public transport) would offer valuable insights into the role of pre-hospital care in ARDS outcomes.

Unfortunately, due to the retrospective design of our study and limitations in medical record documentation, detailed information on the specific mode of transport was not consistently available. Among the 353 patients included in the study, transport data were available for only 214 individuals. Even within this subset, the classification of transport type was often incomplete or inconsistently recorded, precluding a reliable stratified analysis by transport modality. We acknowledge that this limitation may impact the depth of our conclusions regarding the effect of inter-hospital transport on patient outcomes. Specifically, the absence of detailed transport data may have introduced selection bias in our multivariable logistic regression model, potentially leading to the underrepresentation of important pre-hospital factors such as transport duration, level of care during transfer, and type of vehicle used. These variables may influence both the severity of illness upon arrival and the subsequent risk of mortality. To address this limitation, we have expanded the Limitations section of the revised manuscript to explicitly acknowledge the potential impact of missing transport data on the robustness of our findings (Page 35, Lines 632-638). Despite these constraints, our analysis still revealed a critical and actionable finding: the use of endotracheal intubation during transport was independently associated with reduced hospital mortality, underscoring the importance of ensuring adequate respiratory support during inter-facility transfers, regardless of the type of transport.

We appreciate the opportunity to clarify this aspect of our study and believe that the revised discussion more accurately reflects both the strengths and limitations of our transport-related findings.

5. Cytokine Adsorption Therapy:

- Include a brief discussion on the potential impact of cytokine adsorption therapy on outcomes. Even if no formal analysis was done, acknowledging its role and the limitations of this therapy would add valuable context.

Our answer:

Thank you very much for your thoughtful and constructive suggestion. A brief discussion of the potential impact of cytokine adsorption therapy (ECAT) would provide essential context, even in the absence of a formal efficacy analysis.

In response, we have revised and expanded the Discussion section to include a more nuanced interpretation of the role of ECAT in ARDS management and its limitations within our study. The revised text now reads:

“ARDS is increasingly recognized as a syndrome of systemic immune dysregulation. Although the concept of a cytokine storm was initially suggested,(76) later studies have indicated that systemic cytokine levels in ARDS patients are generally lower than previously thought, except in severe COVID-19 cases(26, 77). The use of ECAT has been explored as a potential intervention for immune modulation. While detailed data on ECAT were not available in this study, the use of ECAT was associated with an increased risk of hospital mortality (S14 Table as shown in S1 File), likely reflecting its role as a rescue therapy

---

## [Decision Letter · Decision Letter 2]

30 Sep 2025

Dear Dr. Do,

Thank you for submitting your manuscript to PLOS ONE. After careful consideration, we feel that it has merit but does not fully meet PLOS ONE’s publication criteria as it currently stands. Therefore, we invite you to submit a revised version of the manuscript that addresses the points raised during the review process.

Thank you for submitting the revised version of your manuscript entitled Factors related to mortality in patients with acute respiratory distress syndrome (ARDS) in a lower middle-income country: a retrospective observational study to *PLOS ONE* . We appreciate the substantial effort you have made to address the previous reviewer and editorial comments. Both reviewers acknowledge that the manuscript has improved considerably, with clearer presentation, stronger methodology, and an extended dataset that enhances the study’s relevance. The focus on inter-hospital transport and pre-hospital care adds an important dimension that is often underrepresented in the ARDS literature, particularly in resource-limited settings.

After careful consideration of the reviewers’ feedback, the editorial decision is: **Minor Revision** .

While the manuscript is close to being suitable for publication, several refinements are still recommended to strengthen the clarity and impact of your work:

**Mortality Rate Contextualization**Please provide a deeper discussion of the high observed mortality rate (61.5%), particularly in relation to referral bias, resource limitations, and the availability of advanced therapies.**COVID-19 Subgroup Analysis**Expand the discussion to more explicitly address how the pandemic may have shaped ARDS case-mix, transport practices, and access to ventilatory support.**Transport Data Limitations**Highlight the limitations of the transport-related data more clearly, and if possible, provide a conceptual framework to guide interpretation of these findings.**Adjunctive Therapies**A brief discussion on adjunctive therapies (e.g., prone positioning, ECMO, HFNC, cytokine adsorption) would strengthen completeness, even if data remain limited.**Language and Style**We encourage further polishing of the language, including shortening long sentences, ensuring abbreviation consistency, and improving overall readability.

Please revise the manuscript accordingly and submit the revised version together with a detailed response letter indicating how each point has been addressed.

We look forward to receiving your revised manuscript.

Kind regards,

Gurmeet Singh, M.D., Ph.D.,

Academic Editor

PLOS ONE

Journal Requirements:

**Additional Editor Comments:**

Dear Authors,

Thank you for submitting the revised version of your manuscript entitled "Factors related to mortality in patients with acute respiratory distress syndrome (ARDS) in a lower middle-income country: a retrospective observational study" to Plos One. We appreciate the substantial effort you have made to address the previous reviewer and editorial comments. Both reviewers acknowledge that the manuscript has improved considerably, with clearer presentation, stronger methodology, and an extended dataset that enhances the study’s relevance. The focus on inter-hospital transport and pre-hospital care adds an important dimension that is often underrepresented in the ARDS literature, particularly in resource-limited settings.

After careful consideration of the reviewers’ feedback, the editorial decision is: Minor Revision.

While the manuscript is close to being suitable for publication, several refinements are still recommended to strengthen the clarity and impact of your work:

Mortality Rate Contextualization

Please provide a deeper discussion of the high observed mortality rate (61.5%), particularly in relation to referral bias, resource limitations, and the availability of advanced therapies.

COVID-19 Subgroup Analysis

Expand the discussion to more explicitly address how the pandemic may have shaped ARDS case-mix, transport practices, and access to ventilatory support.

Transport Data Limitations

Highlight the limitations of the transport-related data more clearly, and if possible, provide a conceptual framework to guide interpretation of these findings.

Adjunctive Therapies

A brief discussion on adjunctive therapies (e.g., prone positioning, ECMO, HFNC, cytokine adsorption) would strengthen completeness, even if data remain limited.

Language and Style

We encourage further polishing of the language, including shortening long sentences, ensuring abbreviation consistency, and improving overall readability.

Please revise the manuscript accordingly and submit the revised version together with a detailed response letter indicating how each point has been addressed.

We look forward to receiving your revised manuscript soon.

Reviewers' comments:

Reviewer's Responses to Questions

**Comments to the Author**

Reviewer #2: All comments have been addressed

Reviewer #3: All comments have been addressed

2. Is the manuscript technically sound, and do the data support the conclusions?

Reviewer #2: Yes

Reviewer #3: Yes

3. Has the statistical analysis been performed appropriately and rigorously?

Reviewer #2: Yes

Reviewer #3: Yes

4. Have the authors made all data underlying the findings in their manuscript fully available?

Reviewer #2: Yes

Reviewer #3: Yes

5. Is the manuscript presented in an intelligible fashion and written in standard English?

Reviewer #2: Yes

Reviewer #3: Yes

Reviewer #2: You guys did a solid job with the revisions. The paper is clearer, more rigorous, and adds useful insights into ARDS outcomes in a resource-limited setting. The focus on transport and pre-hospital care makes it stand out, since most ARDS studies focus only on ICU interventions.

Main Points:

They addressed all prior reviewer/editor concerns. The issue of “dual publication” is clarified—this is an updated dataset with a new focus, not a duplicate.

You added a COVID vs. pre-COVID analysis, which showed no major difference in mortality but gives extra context.

The stats section is much stronger now. You explained how they built the model, checked for multicollinearity (all fine), and discussed missing data honestly.

Key result: intubation during transport strongly predicts survival. The effect size is big, but given the wide confidence intervals, I’d interpret it cautiously. Still, it’s a meaningful and practice-relevant finding.

You added discussion of cytokine adsorption therapy—helpful for context, while noting it was used mainly in very sick patients.

Reviewer #3: The manuscript has been substantially improved compared to the earlier version, and the authors have clearly made an effort to address the reviewers’ and editor’s concerns. The extended dataset, inclusion of the pandemic period, and focus on inter-hospital transport strengthen the study and add relevance. The finding regarding endotracheal intubation during transfer remains a novel and clinically meaningful contribution.

Some areas would still benefit from refinement:

- The high mortality rate (61.5%) needs deeper contextualization, particularly in relation to referral bias, resource limitations, and lack of advanced therapies.

- The COVID-19 subgroup analysis is useful, but the discussion could emphasize more explicitly how the pandemic may have shaped ARDS case-mix, transport practices, and access to ventilatory support.

- The limitations regarding transport data should be highlighted further; if possible, provide a conceptual framework to guide interpretation.

- A short discussion on adjunctive therapies (cytokine adsorption, prone positioning, ECMO, HFNC) would strengthen the completeness of the discussion, even if data are limited.

- Minor polishing of language (shorter sentences, abbreviation consistency) would improve readability.

Overall, this revision represents a meaningful contribution to the ARDS literature in resource-limited settings. With the suggested refinements, it should be suitable for publication.

**Do you want your identity to be public for this peer review?** For information about this choice, including consent withdrawal, please see our Privacy Policy

Reviewer #2: **Yes: ** Sagar Bathija

Reviewer #3: No

---

## [Author Response · Author response to Decision Letter 3]

3 Nov 2025

Dr. Gurmeet Singh, M.D., Ph.D.

Academic Editor

PLOS ONE

November 3, 2025

Dear Dr. Gurmeet Singh,

On behalf of all the authors, we are resubmitting our revised manuscript titled “Factors related to mortality in patients with acute respiratory distress syndrome (ARDS) in a lower middle-income country: a retrospective observational study” (PONE-D-24-35750R2).

We sincerely appreciate the thoughtful comments and suggestions provided by the Editors and Reviewers. We have carefully addressed all feedback and revised our manuscript accordingly. These insights have significantly improved the quality of our work, and we hope the Editor finds our revised manuscript suitable for publication in PLOS ONE.

We confirm that this research is original, has not been published elsewhere, and is not currently under consideration by any other journal. All authors have read and approved the manuscript, have agreed on the order of authorship, and hold the necessary permissions and rights for the data presented.

Below, we have included our point-by-point responses to the comments from the Editor and the Reviewers. Thank you for considering this submission.

Sincerely,

Chinh Quoc Luong, MD, PhD

Neuro Intensive Care Department,

Neurology Center, Bach Mai Hospital,

No. 78, Giai Phong, Dong Da district,

Hanoi 100000, Vietnam

Email: luongquocchinh@gmail.com

Son Ngoc Do, MD, PhD

Center for Critical Care Medicine,

Bach Mai Hospital,

No. 78, Giai Phong Road, Dong Da District,

Hanoi, 100000, Vietnam.

Email: sonngocdo@gmail.com

*****

We thank the Editors and the Reviewers for the valuable comments and suggestions that have helped us improve the contents of this paper. In what follows, we have used the boldface to indicate the Editors' and the Reviewers' comments and the standard font face for our responses. We also highlighted in yellow the modifications that we performed on the manuscript.

RESPONSE TO EDITOR COMMENTS

PONE-D-24-35750R2

Factors related to mortality in patients with acute respiratory distress syndrome (ARDS) in a lower middle-income country: a retrospective observational study

PLOS ONE

Dear Dr. Do,

Thank you for submitting your manuscript to PLOS ONE. After careful consideration, we feel that it has merit but does not fully meet PLOS ONE’s publication criteria as it currently stands. Therefore, we invite you to submit a revised version of the manuscript that addresses the points raised during the review process.

Our answer:

Thank you for your encouraging feedback and for the opportunity to revise our manuscript.

We sincerely appreciate the time and effort that you and the reviewers have devoted to evaluating our work. In response to the thoughtful comments and suggestions provided, we have carefully revised the manuscript to address all points raised during the review process. These changes have strengthened the clarity, rigor, and overall quality of our study.

We look forward to your continued evaluation and hope the revised version meets the journal’s publication criteria.

ACADEMIC EDITOR:

Thank you for submitting the revised version of your manuscript entitled "Factors related to mortality in patients with acute respiratory distress syndrome (ARDS) in a lower middle-income country: a retrospective observational study" to PLOS ONE. We appreciate the substantial effort you have made to address the previous reviewer and editorial comments. Both reviewers acknowledge that the manuscript has improved considerably, with clearer presentation, stronger methodology, and an extended dataset that enhances the study’s relevance. The focus on inter-hospital transport and pre-hospital care adds an important dimension that is often underrepresented in the ARDS literature, particularly in resource-limited settings.

After careful consideration of the reviewers’ feedback, the editorial decision is: Minor Revision.

Our answer:

Thank you for your thoughtful and encouraging feedback regarding our revised manuscript.

We are grateful to you and the reviewers for recognizing the improvements in clarity, methodology, and dataset relevance. We appreciate your acknowledgment of the added focus on inter-hospital transport and pre-hospital care, which makes a meaningful contribution to the literature on acute respiratory distress syndrome (ARDS) in resource-limited settings.

We will carefully address the remaining points raised and submit the minor revision promptly. Thank you again for your continued guidance and support throughout the review process.

While the manuscript is close to being suitable for publication, several refinements are still recommended to strengthen the clarity and impact of your work:

Mortality Rate Contextualization

Please provide a deeper discussion of the high observed mortality rate (61.5%), particularly in relation to referral bias, resource limitations, and the availability of advanced therapies.

Our answer:

We sincerely thank the Academic Editor for this important suggestion. In the revised manuscript, we have expanded the discussion to provide deeper contextualization of the high mortality rate. Specifically, we have now highlighted the potential impact of referral bias resulting from the transfer of more severe cases from local hospitals, the limitations in critical care resources at both the local and central levels, and the restricted availability of advanced therapies, such as ECMO and high-flow nasal cannula (HFNC). We have explicitly discussed these factors to better explain the observed mortality rate in comparison with international cohorts, as follows:

“A key factor in our study is the selective nature of our cohort, as most ARDS patients were transferred from local hospitals to a central tertiary facility (Table 1), introducing significant referral bias. Patients are typically transferred only after initial management fails at lower-level facilities, resulting in a cohort with more advanced disease severity upon arrival. In Vietnam, patients with ARDS are often initially diagnosed with severe pneumonia at local hospitals and transferred only when their condition worsens, delaying diagnosis and treatment and potentially contributing to higher mortality (26, 27, 31, 32). Resource limitations further compound this issue. Vietnam's healthcare system faces chronic underfunding, shortages of trained intensivists, advanced diagnostic equipment, and critical care beds, particularly at provincial and district levels (levels II and III per the Ministry of Health, (29). These constraints lead to suboptimal early interventions, such as delayed initiation of lung-protective ventilation or adjunctive therapies, which are more readily available in HICs.

Pneumonia was the predominant etiology of ARDS in our study (Table 3), aligning with prior studies such as VALID (19.3%; 125/646, (8), ALIEN (42.4%; 108/255, (6), and LUNG SAFE (59.4%; 1794/3022, (9), though our cohort exhibited a higher prevalence. This predominance likely exacerbates mortality, as pneumonia and sepsis are leading causes of early death in ARDS, alongside MODS (12-15). In the present study, the SOFA score (Table 4) is comparable to those in LUNG SAFE (mean 10.1 [95% CI: 9.9–10.2], (9), but our PaO2/FiO2 ratio is lower (Table 4 vs. 161 mmHg [95% CI: 158–163] in LUNG SAFE, (9), suggesting greater hypoxemia severity. A large observational study (n=3670) also indicates that mortality rises with ARDS severity (19, 36), potentially explaining our elevated mortality rates due to the high pneumonia burden. Additionally, the lack of widespread access to advanced therapies, such as ECMO (utilized in only 8.5% of our cohort; Table 5) and HFNC during transport or early management (29, 30, 55), limits rescue options for refractory cases. In contrast to HICs, where ECMO is more routinely available and associated with better outcomes in high-volume centers (56, 57), LMICs like Vietnam face barriers including high costs, limited expertise, infrastructure gaps, and global disparities in access. These factors collectively contribute to the higher mortality observed here compared to international cohorts (e.g., 23.7% in VALID (8) and 40.1% in LUNG SAFE (9)). To reduce ARDS mortality in Vietnam, enhancing local healthcare resources (e.g., human expertise, medical equipment, and infrastructure) is critical, as evidenced by ongoing efforts to upgrade hospitals and address workforce shortages.” (Lines 519-550, Pages31-32)

COVID-19 Subgroup Analysis

Expand the discussion to more explicitly address how the pandemic may have shaped ARDS case-mix, transport practices, and access to ventilatory support.

Our answer:

We thank the Academic Editor for this valuable recommendation. We have revised the discussion to more explicitly address how the COVID-19 pandemic influenced ARDS case-mix, transport practices, and access to ventilatory support. We now describe how infection control measures, resource constraints, and evolving treatment protocols during the pandemic altered referral patterns, increased the use of endotracheal intubation during transport, and reduced the availability of invasive ventilation at local hospitals. These contextual factors are now integrated into the interpretation of our COVID-19 subgroup findings, as follows:

“During the pandemic, the ARDS case-mix shifted significantly, with a marked decline in cases attributed to seasonal viruses, such as influenza A (H1N1; Table 7). This decline was due to non-pharmaceutical interventions (NPIs) such as mask-wearing and social distancing (87), mirroring global trends (87, 88), which contrasts with pre-pandemic data in Vietnam, where influenza was a primary cause of ARDS, underscoring the need for robust epidemiological surveillance to monitor such changes. Conversely, there was an influx of ARDS cases associated with SARS-CoV-2, particularly the Delta variant, as critically ill COVID-19 patients often faced delayed hospital admissions, leading to severe pneumonia, rapid ARDS progression, and high mortality (25, 78). This etiological shift resulted in a more homogeneous viral-driven ARDS cohort during the pandemic, which differed from the diverse pre-pandemic mix of bacterial and other viral pneumonias, and potentially influenced treatment protocols in resource-limited settings. There was also a significant reduction in the use of both noninvasive and invasive MV at referring hospitals (Table 7). This decline stemmed from resource constraints (29), including acute ventilator shortages in LMICs that restricted MV access, and infection control measures to limit aerosol-generating procedures, aligning with reports of strained healthcare systems worldwide (78, 88, 89). Such limitations exacerbated challenges in early respiratory support, prompting greater reliance on alternatives, such as HFNC, when available (90). Transport practices evolved as reliance on ET use increased during inter-hospital transfers to tertiary centers (Table 7), reflecting adaptations to manage infectious risks, stabilize airways in deteriorating patients, and cope with overwhelmed local facilities amid surges. This trend, observed in other LMICs (91, 92), often led to increased transfers for critical cases due to uneven resource distribution. Still, it also added logistical and psychological burdens on patients and providers (93, 94). Vietnam's shortages of ventilators and critical care infrastructure were key drivers of these changes. A notable shift in ARDS management was the reduced use of invasive MV on the first day of admission during the pandemic (Table 7). This change is consistent with evolving guidelines favoring noninvasive strategies, such as HFNC, for COVID-19-related ARDS to conserve ventilators and minimize intubation risks (57, 90). Pre-pandemic, invasive MV was standard (19, 36); this change arose from both resource constraints and emerging evidence (57), underscoring the need for context-specific protocols in low-resource settings. Despite these shifts, hospital mortality rates for ARDS remained stable between pre-pandemic and pandemic periods (Table 7), suggesting resilience in Vietnam's critical care system. However, rates were consistently high, exceeding the 35–45% reported in HICs (9), likely due to limited infrastructure, including shortages of trained intensivists, ventilators, and standardized protocols.” (Lines 650-684, Pages36-37)

Transport Data Limitations

Highlight the limitations of the transport-related data more clearly, and if possible, provide a conceptual framework to guide interpretation of these findings.

Our answer:

We appreciate this thoughtful comment. In the revised limitations section, we have expanded our discussion of the transport data, noting the incomplete documentation of transfer duration, level of care provided en route, and vehicle type. To guide interpretation, we have added a conceptual framework that emphasizes how deficiencies in pre-hospital and inter-hospital systems—such as a lack of standardization, limited monitoring, and staffing variability—may have influenced outcomes. This framework helps contextualize the observed association between endotracheal intubation during transport and reduced mortality. Further details as follows:

“This study has several limitations that warrant careful consideration when interpreting the findings. First, the retrospective design limited data availability and completeness across key variables (S17 Table in S1 File). Transport-related information was available for only 214 patients and lacked critical details, such as transfer duration, en route care level, vehicle type, and interventions. These gaps introduce potential selection bias into the multivariable logistic regression analyses, as unmeasured confounders, such as transfer delays or variations in monitoring, could affect patients' arrival condition and hospital mortality, which undermines model robustness, potentially leading to over- or under-estimation of associations and unreliable inferences about transport's role in ARDS outcomes. Vietnam's pre- and inter-hospital transfer systems remain underdeveloped, lacking standardization, integration, and regulatory oversight (30, 32, 34, 55, 95), which hampers data collection for surveillance, quality improvement, and research and limits the generalizability of our findings to other LMICs with similar constraints. To interpret these transport limitations, we adapt Donabedian's healthcare quality framework, categorizing factors into structural, process, and outcome dimensions, thereby highlighting how deficiencies in LMIC inter-hospital transport for ARDS patients amplify risks, unlike in HICs with specialized retrieval teams and protocols. Structural quality (i) involves foundational resources, such as ambulance types (hospital-operated vs. private/informal), staffing (trained physicians vs. nurses/bystanders), and equipment (ventilators, oxygen, ECMO). Incomplete documentation in our study underscores LMIC challenges, such as resource scarcity, potentially underestimating risks from equipment failure or inadequate stabilization. Process quality (ii) covers protocols and actions, including pre-transfer stabilization, communication, and interventions like ET or MV. Our data showed limited ET use (60.1% of transported patients), possibly delaying respiratory support and worsening hypoxemia; non-standardized processes may bias toward poorer outcomes. Outcome quality (iii) includes transport events (e.g., hypoxia, hypotension) and metrics like hospital mortality. ET use during transport was independently associated with lower mortality (Table 8; S14 Table in S1 File), but missing details hinder causal inferences due to unadjusted confounders. Despite these issues, our findings suggest prioritizing ET could mitigate transfer risks. Future prospective studies should use standardized protocols, including mobile ECMO or ET/MV interventions, to optimize ARDS transport in LMICs.” (Lines 685-715, Pages 37-38)

Adjunctive Therapies

A brief discussion on adjunctive therapies (e.g., prone positioning, ECMO, HFNC, cytokine adsorption) would strengthen completeness, even if data remain limited.

Our answer:

We thank the Academic Editor for this excellent suggestion. We have added a dedicated section in the discussion addressing adjunctive

---

## [Editor Report · Decision Letter 3]

4 Nov 2025

Factors related to mortality in patients with acute respiratory distress syndrome (ARDS) in a lower middle-income country: a retrospective observational study

PONE-D-24-35750R3

Dear Dr. Do,

We’re pleased to inform you that your manuscript has been judged scientifically suitable for publication and will be formally accepted for publication once it meets all outstanding technical requirements.

Kind regards,

Gurmeet Singh, M.D., Ph.D.,

Academic Editor

PLOS ONE
---

## [Editor Report · Acceptance letter]

PONE-D-24-35750R3

PLOS ONE

Dear Dr. Do,

I'm pleased to inform you that your manuscript has been deemed suitable for publication in PLOS ONE. Congratulations! Your manuscript is now being handed over to our production team.

Kind regards,

on behalf of

Dr. Gurmeet Singh

Academic Editor

PLOS ONE